# DitHub: A Modular Framework for Incremental Open-Vocabulary Object Detection

**Chiara Cappellino**        **Gianluca Mancusi**        **Matteo Mosconi**

**Angelo Porrello**        **Simone Calderara**        **Rita Cucchiara**

AImageLab - University of Modena and Reggio Emilia
`name.surname@unimore.it`

## Abstract

Open-Vocabulary object detectors can generalize to an unrestricted set of categories through simple textual prompting. However, adapting these models to rare classes or reinforcing their abilities on multiple specialized domains remains essential. While recent methods rely on monolithic adaptation strategies with a single set of weights, we embrace modular deep learning. We introduce DitHub, a framework designed to build and maintain a library of efficient adaptation modules. Inspired by Version Control Systems, DitHub manages expert modules as branches that can be fetched and merged as needed. This modular approach allows us to conduct an in-depth exploration of the compositional properties of adaptation modules, marking the first such study in Object Detection. Our method achieves state-of-the-art performance on the ODinW-13 benchmark and ODinW-O, a newly introduced benchmark designed to assess class reappearance.

## 1   Introduction

Object detection is a cornerstone of computer vision, with recent advancements [48, 2, 57, 32, 24] enabling architectures that integrate Vision and Language large-scale pre-training [25]. This has given rise to *Open-Vocabulary Object Detection*, where models can classify and localize objects beyond a predefined set of categories. The ability to process arbitrary textual queries and detect corresponding objects has profound implications for real-world applications, including medical imaging [18], safety and security [41], and industrial quality control [38]. For instance, the ability to detect rare or atypical objects from a safety camera is crucial for issuing early warnings about uncommon dangers. However, this requires Open-Vocabulary detectors to adapt effectively, enabling them to recognize categories beyond their pre-training knowledge distribution.

Recent approaches, such as those detailed in [4], have extended Vision-Language detectors to **incrementally** accommodate new categories while preserving robust capabilities [40]. However, these methods employ *monolithic* adaptation, where all newly acquired knowledge is condensed into a single set of weights. This approach presents challenges in real-world scenarios that demand updates to specific concepts which may reappear under varied input modalities. For instance, in safety and security applications, the class "person" may need to be detected not only in standard RGB imagery but also in thermal imagery, necessitating continuous model adaptation to ensure consistent performance across different data types. Additionally, for rare or complex categories, incremental adaptation is crucial to refine the base model for fine-grained concepts. In such scenarios, a monolithic deep architecture faces challenges similar to managing a complex program written on

39th Conference on Neural Information Processing Systems (NeurIPS 2025).

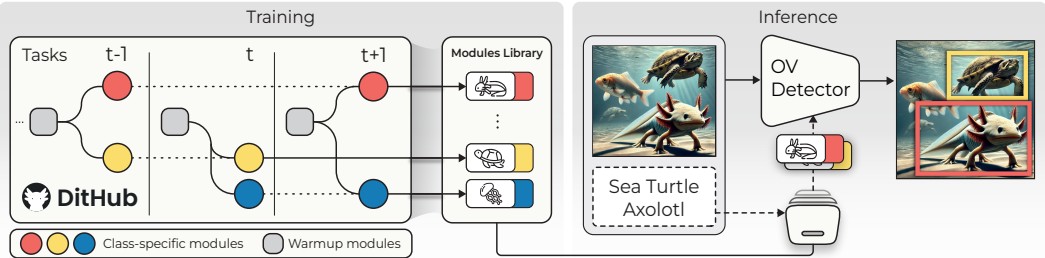

Figure 1: DitHub: a novel system to share and update efficient adaptation modules for Open-Vocabulary Object Detection.[1]

a single page of code. Namely, when treating multiple classes within a unique set of weights, it becomes hard to update specific concepts selectively, without compromising information related to other concepts. Considering less-represented categories, the corresponding concepts can become blurred within the unique set of weights, leading to potential degradation in performance over time.

We advocate for **modular** approaches, as outlined in Modular Deep Learning [36], which reimagine neural networks as adaptable and composable entities rather than rigid, monolithic structures. By maintaining a library of expert modules, neural architectures can achieve class-specific specialization, activating only the relevant module when needed. Moreover, this modular design enables selective updates and seamless composability, allowing class-specific modules trained in different domains to be efficiently merged. Finally, modularity offers a practical solution to mitigate biases — if a module exhibits undesirable biases toward a particular category, it can be selectively removed or adjusted without disrupting the entire model, ensuring more precise and fair adaptation.

With modularity at its core, we introduce **DitHub** (where D stands for Detection), a framework inspired by Version Control Systems that efficiently manages a growing library of detection modules. This library expands incrementally as new groups of categories, referred to as tasks [47], are introduced. Upon encountering a new task, DitHub initializes an efficient module (*e.g.*, LoRA [13]), which first undergoes a broad pre-tuning phase across all classes of the task, referred to as the **warmup** phase. This stage establishes a solid and class-agnostic foundation, enabling the subsequent **specialization** of the module for specific classes. Following the warmup phase, the module `branches` into specialized experts, each optimized independently. To enable selective updates for classes previously introduced to the model, DitHub first performs a `fetch` operation to access potentially existing class-specific modules. If these exist, instead of starting the training from the warmup, DitHub uses a `merged` initialization — combining the retrieved pre-existing class-specific modules with the current warmup module. This merged initialization facilitates smooth domain knowledge transfer and ensures continuity of class-specific learning across tasks.

Moreover, to ensure scalability with respect to the number of classes, DitHub **shares** a subset of parameters across different adaptation modules, enhancing memory efficiency without compromising individual class adaptability. At inference, it enables the selective activation of specialized modules for rare or complex classes, ensuring robust adaptation while maintaining the backbone's zero-shot and Open-Vocabulary capabilities. A high-level overview of our approach is depicted in Figure 1.

Following [4], we evaluate DitHub on the ODinW-13 [20] benchmark, where it outperforms the state-of-the-art by a substantial +4.21 mAP in the Incremental Vision-Language Object Detection setting. We also introduce ODinW-O (Overlapped), a curated subset of ODinW-35 [19] focusing on classes **re-appearing** across multiple tasks. Furthermore, we thoroughly investigate and ablate the compositional capabilities of our modular approach, marking the first such exploration in Object Detection. In summary, our key contributions are:

- We investigate the importance of decoupling a warmup phase from a specialization one in the context of efficient adaptation modules.
- We introduce DitHub, a framework inspired by Version Control Systems, designed to manage a growing library of efficient modules for Open-Vocabulary detection.

---

[1]The creature in the red box is an axolotl, symbolizing rarity as a deviation from pre-training. We use a stylized axolotl as the icon for DitHub.

- We achieve state-of-the-art performance on ODinW-13 and the newly introduced ODinW-O.
- We conduct an in-depth analysis of the compositional and specialization capabilities of class-specific modules, a first in the context of Object Detection.

## 2   Related Works

**Incremental Vision-Language Object Detection.** Vision-Language Object Detection (VLOD), also referred to as Open-Vocabulary detection [4], is the task of detecting and recognizing objects beyond a fixed set of predefined categories. Early approaches leverage powerful visual and textual encoders (*e.g.*, CLIP [40]) to enable object detectors to recognize virtually any object specified by a textual query [10, 55]. A fundamental trend in this field has been the reliance on large-scale pre-training on extensive datasets [43], which has become standard practice for tasks such as classification. In the context of Vision-Language Object Detection, state-of-the-art methods have followed this paradigm, with GLIP [20] being one of the first notable examples. GLIP reframes Object Detection as a phrase-grounding problem, leading to a pre-trained model capable of generalizing to unseen objects by integrating textual and visual semantics from large-scale datasets. Building on GLIP's problem formulation, Grounding DINO [25] extends this paradigm by incorporating merged visual and textual semantics at multiple network stages.

Recently, Incremental Vision-Language Object Detection (IVLOD) [4] has emerged as a natural extension of the VLOD paradigm. By incorporating Incremental Learning [47, 17, 42, 51], IVLOD addresses the challenge of fine-tuning a pre-trained Vision-Language model on specific object categories while preserving its zero-shot [35] capabilities. Following [4], we adopt Grounding DINO as the backbone and use IVLOD to explore the merging of class-specific modules.

**Parameter Efficient Fine-Tuning and Model Merging.** The rise of Transformer models [48, 7] has made pre-training a standard approach in machine learning. Parameter-Efficient Fine-Tuning (PEFT) methods, such as Prefix Tuning [21], Adapters [12], and LoRA [13], adapt pre-trained models to downstream tasks by updating only a small fraction of parameters while keeping the rest frozen.

In the context of IVLOD, ZiRa [4] introduces a distinct approach that can be viewed as a specialized form of parallel adapters [28], incorporating reparameterizable side branches. In contrast, our work adopts LoRA and explores the potential of merging class-specific LoRA modules.

Several studies have investigated LoRA module fusion. An early approach relies on simple arithmetic operations such as addition and subtraction [56]. More advanced techniques focus on learning optimal combination coefficients, determining the contribution of each LoRA module in the merging process [14, 53, 52, 29, 54]. Recently, researchers have introduced closed-form solutions for LoRA merging [44] and proposed regularization objectives to ensure that the fused modules remain close to the pre-trained weight distribution [37]. In this respect, our approach takes a novel direction, being the first to study module merging in the context of Object Detection.

## 3   Preliminaries

**Problem setting.** We consider a sequence of tasks $\{1, \ldots, T\}$, where each task $t$ corresponds to a dataset $\mathcal{D}_t$. Each dataset consists of samples $(x, y)$, where $x$ represents the input data, and $y$ contains the associated labels, including ground truth bounding boxes.

Given a pre-trained Vision-Language detector $f(\cdot; \Theta)$, parameterized by $\Theta$, the objective of *Incremental Vision-Language Object Detection* (IVLOD) is to improve performance on newly introduced tasks while preserving original zero-shot capabilities. Formally, for a given task $t$, we seek an adaptation $\Delta$ that minimizes the following objective:

$$\min_{\Delta} \mathbb{E}_{(x,y) \sim \mathcal{D}_t} \ell(f(x; \Theta + \Delta), y) + \ell_{\text{zero-shot}}, \tag{1}$$

where $\ell$ is the Object Detection loss function. The term $\Theta + \Delta$ represents the set of adapted model parameters. Additionally, $\ell_{\text{zero-shot}}$ quantifies the performance difference in the zero-shot setting before and after adaptation.

Each task $t$ introduces a set of classes $\mathcal{C}_t$. Unlike standard Incremental Learning [47, 51, 50, 46], IVLOD allows class overlap between tasks. In other words, there exist task pairs $(t, t')$, with $t' \neq t$, such that $\mathcal{C}_t \cap \mathcal{C}_{t'} \neq \emptyset$.

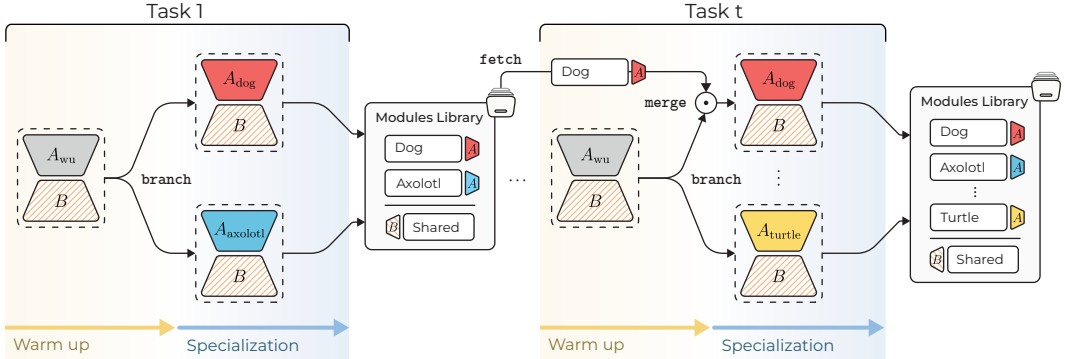

Figure 2: Overview of DitHub. Through three primitives – *i.e.*, `branch`, `fetch`, `merge` – DitHub allows to update a library of class-specific modules for Open-Vocabulary Object Detection.

**LoRA.** Low-Rank Adaptation [13] is a technique to significantly reduce the number of trainable parameters. Given a generic layer with weight matrix $W \in \mathbb{R}^{d \times k}$, LoRA introduces low-rank matrices $A \in \mathbb{R}^{r \times k}$ and $B \in \mathbb{R}^{d \times r}$ — with $r \ll \min(d, k)$ — such that the adapted weight $W'$ is:

$$W' = W + \Delta W, \quad \text{where} \quad \Delta W = BA. \tag{2}$$

The use of these low-rank matrices significantly reduces memory usage and computational overhead, making LoRA an efficient alternative to fine-tuning the entire model.

## 4 DitHub

Version Control Systems are ubiquitous tools used to manage and track changes in code repositories. Inspired by this, we propose DitHub — a framework that operates akin to a Version Control System, enabling the creation and expansion of a library of efficient adaptation modules for Vision-Language object detectors. DitHub develops and maintains class-specific *expert* modules using a simple yet effective strategy that requires no architectural modifications or additional loss terms (Section 4.1).

Following [4], we consider the pre-trained Grounding DINO as the base architecture for Open-Vocabulary detection and use LoRA for fine-tuning. We propose an efficient pipeline that handles LoRA's low-rank matrices with two objectives: matrix $A$ encodes class-specific knowledge, while matrix $B$ captures general knowledge and is shared among tasks to maintain overall *efficiency* (Section 4.2). Finally, when fine-grained adaptation is needed (*e.g.*, detecting a rare object within a scene), DitHub enables the dynamic extraction of the corresponding expert modules, ensuring precise and efficient inference, as outlined in Section 4.3. The pseudocode of DitHub is provided in Algorithm 1.

### 4.1 Training Class-Specific Modular Detectors

To manage a library of class-specific modules, our approach is to train a tailored $A$ matrix for each class to enable specialization. In contrast, the $B$ matrix is shared across classes and continuously fine-tuned as tasks progress. Thus, at task $t$, we train a total of $|\mathcal{C}_t|+1$ matrices per adaptable layer — one for each class and one shared $B$ matrix.

However, in Object Detection, creating class-specific experts is non-trivial, as images often contain multiple objects from different categories, making it difficult to isolate a single class. To tackle this, we adopt a **stochastic training strategy** for specializing the $A$ matrices: for each image, we randomly select one of the present classes and update its corresponding $A$ matrix. However, preliminary experiments show this approach is ineffective for rare categories, as random selection limits their exposure to sufficient training data.

As better highlighted in Algorithm 1, we overcome this issue by introducing an earlier **warmup** phase at the beginning of each task. This phase is meant to provide a robust, class-agnostic initialization, ensuring that the subsequent **specialization** stage is more effective.

| Algorithm 1: DitHub Training | 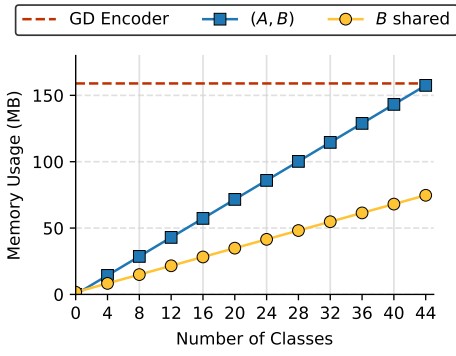 |
|---|---|

**Input:** $T$ tasks each containing $|\mathcal{C}_t|$ classes; library $\mathcal{A} \leftarrow \{\}$ of class experts; global $B$.

**for** task $t \in \{1, \dots, T\}$ **do**
  $(A_{\text{wu}}, B^{\text{opt}}) \leftarrow$ **warmup** *(training)*
  **for** class $c \in \mathcal{C}_t$ **do**            ▷ Branch
    **if** $A_c \in \mathcal{A}$ **then**              ▷ Fetch
      $A_c^{\text{cur}} \leftarrow$ use Equation 3      ▷ Merge
    **else**
      $A_c^{\text{cur}} \leftarrow A_{\text{wu}}$
    $(A_c, B^{\text{opt}}) \leftarrow$ **specialization** *(training)*
    $\mathcal{A} \leftarrow \mathcal{A} \cup A_c$
  $B \leftarrow$ use Equation 4

Figure 3: Memory requirements as the number of classes grows.

**Warmup.** At the onset of the new task, a shared warmup matrix $A_{\text{wu}}$ is instantiated and optimized (see Figure 2, yellow arrow) to capture class-agnostic yet task-specific knowledge. This phase, which runs for a fixed number of epochs, provides a common initialization for the $A$ modules before their specialization. By doing so, the specialized modules start from a common base point, a factor that has been shown to greatly enhance model compositionality [33, 39]. Often associated with improved linear mode connectivity between models [9], these properties are leveraged by our approach to combine class-specific modules based on input textual queries, as detailed in Section 4.3. Moreover, considering rare and underrepresented classes with a few training examples, the warmup phase seeks to establish an already robust and effective initialization.

**Specialization.** After the warmup phase, we proceed to the core of DitHub: the specialization phase. To train class-specific modules, DitHub generates $|\mathcal{C}_t|$ distinct branches (`branch` in Figure 2), each dedicated to a single class. These branches undergo independent optimization, represented by a blue arrow in Figure 2, to capture class-specific capabilities. In doing so, for each training image, we employ the stochastic strategy introduced in Section 4.1 to select a single expert for training.

Notably, the modular design of our approach effectively handles potential *class overlaps* across consecutive tasks. In real-world scenarios, classes may reappear in new domains with altered distributions or semantics. For instance, consider the class "dog", originally learned from real-world camera images, later appearing in a task involving thermal imagery. The goal is to retain prior knowledge and enable a "dog" expert to recognize the entity across domains, despite semantic variation. In this case, the modularity of DitHub enables the selective adaptation of the "dog" module, while minimizing interference with other classes, such as the "axolotl" module.

To favor selective updates, DitHub performs a `fetch` operation (Figure 2), which simply retrieves the class-specific module from previous tasks if the class has already been seen — followed by the `merge` (Figure 2) . For instance, considering the class "dog", the previously stored expert module $A_{\text{dog}}^{\text{old}}$ is fetched and merged with the current task's warmup matrix $A_{\text{wu}}$ following a weighted strategy:

$$A_{\text{dog}}^{\text{cur}} = (1 - \lambda_A) A_{\text{dog}}^{\text{old}} + \lambda_A A_{\text{wu}} \tag{3}$$

Here, $\lambda_A$ is set to a low value, prioritizing consolidated knowledge over new insights from the warmup phase. Details on the impact of $\lambda_A$ are in Section 5.2. Crucially, our merging process creates an on-the-fly initialization that blends historical class-specific knowledge with new domain information, resulting in $A_{\text{dog}}^{\text{cur}}$, which serves as the starting point for the specialization phase.

## 4.2 Bridging Modularity with Memory Efficiency

To keep our library memory-efficient, we maintain a single shared $B$ matrix instead of a distinct $B$ for each class. As shown in Figure 3, using class-specific $(A, B)$ pairs results in a linear increase in memory consumption as the number of classes grows. Sharing the $B$ matrix reduces this requirement by half [2]. Additionally, considering DitHub "$B$ shared" (line ⚬), the memory footprint of our

---
[2]In Section A.2, we present DitHub's results when $B$ is not shared.

approach is consistently lower than the storage needed for the Grounding DINO encoder *alone* (line ▪▪). This makes DitHub's memory demands manageable for practical applications. Moreover, memory efficiency is optimized by fine-tuning a targeted portion of the base architecture (Section A.1).

Due to the shared design, the matrix $B$ is more susceptible to interference and catastrophic forgetting [31]. Consequently, we implement a mechanism that ensures smooth updates as tasks progress. Specifically, at the end of each task $t$, we update the matrix $B$ by merging the previous checkpoint, $B_{t-1}$, with the weights learned after the current task, $B^{\text{opt}}$. This process is formulated as follows:

$$B_t = (1 - \lambda_B)B_{t-1} + \lambda_B B^{\text{opt}}. \tag{4}$$

Unlike Equation 3, the merging coefficient $\lambda_B$ is assigned a higher value, giving greater weight to the latest state. This approach results in a dual strategy for $\lambda_A$ and $\lambda_B$, further detailed in Section 5.2.

### 4.3 Inference with DitHub

With our library of efficient, class-specific modules in place for adapting the Vision-Language detector, DitHub enables two complementary inference strategies. For requests targeting individual classes where fine-grained adaptation is needed, the corresponding expert module can be directly selected and used to detect that class. Conversely, when evaluating on a textual prompt that includes multiple classes, we retrieve the corresponding class-specific modules from the library and merge them by computing an average of their weights to create a single composite adaptation module. This unified module is then used in a **single forward pass** to perform detection over all target classes. In our experimental studies, we present results and discussions that address both scenarios.

## 5 Experimental Studies

### 5.1 Incremental Vision-Language Object Detection

**Evaluation settings.** Following [4], we employ ODinW-13 [20] to assess the model ability to learn new tasks while mitigating forgetting. ODinW-13 includes 13 sub-datasets, spanning from the traditional Pascal VOC [8] to more challenging domains with significant distribution shifts, such as thermal imagery. Also, the IVLOD setting requires the Open-Vocabulary detector to retain strong zero-shot capabilities after fine-tuning, which are evaluated on MS COCO [23].

One issue of ODinW-13 is that most classes are confined to a single task among the 13. However, DitHub is conceived to handle recurring classes across domains, necessitating updates to their respective modules. To better assess incremental detection and module reuse, we introduce **ODinW-O** (Overlapped), a subset of ODinW-35 that is designed to reflect realistic scenarios where classes can reoccur across tasks over time. Further details are provided in Section C.

For comparisons, we employ mean Average Precision (mAP) and compare DitHub against six state-of-the-art methods. CL-DETR [26] and OW-DETR [11] are explicitly designed for Incremental Learning, while TFA [49] specializes in few-shot adaptation. AT [12] has been extended to both incremental and few-shot learning, whereas iDETR [6] natively integrates both aspects. Finally, ZiRa [4], our primary competitor, is specifically tailored for IVLOD.

**Implementation details.** We focus on Grounding DINO and adapt only its encoder network, as further validated in Section A.1. The merging coefficient $\lambda$ for the $A$ matrix is set to 0.3 for ODinW-13 and 0.1 for ODinW-O; for the matrix $B$, we set $\lambda = 0.7$. The choice of $\lambda$ and its underlying rationale are further discussed in Section 5.2. During training, we allocate an equal number of epochs to the warmup and the subsequent specialization phases — see Section E for further details.

**Performance on ODinW-13.** Table 1 presents the IVLOD results, including the average across the 13 tasks (Avg) and zero-shot preservation (ZCOCO). We evaluate performance in both the full-data setting and few-shot scenarios with 1, 5, and 10 shots per class. All baseline results, except those of Grounding DINO, ZiRa, and our method, are taken directly from [4]. Each number represents the average of three runs with different random seeds, with standard deviations provided in Section D.1.

In the full-shot setting, DitHub outperforms ZiRa by a substantial +4.21 mAP and achieves the best results in nine out of thirteen tasks. Notably, on ZCOCO, it surpasses ZiRa by a +0.75 mAP, setting a new state-of-the-art in both incremental and zero-shot retention. While not explicitly designed

Table 1: Comparison of mAP values across ODinW-13 tasks. **Avg** represents the average across the 13 tasks, while zero-shot performance is reported in the **ZCOCO** column. Bold indicates best results.

| Shots | Method | ZCOCO | Avg | Ae | Aq | Co | Eg | Mu | Pa | Pv | Pi | Po | Ra | Sh | Th | Ve |
|---|---|---|---|---|---|---|---|---|---|---|---|---|---|---|---|---|
| 0 | G-Dino | 47.41 | 46.80 | 19.11 | 20.82 | 64.75 | 59.98 | 25.34 | 56.27 | 54.80 | 65.94 | 22.13 | 62.02 | 32.85 | 70.38 | 57.07 |
| 1 | TFA | 18.84 | 39.50 | 18.25 | 15.81 | 63.90 | 50.79 | 28.47 | 50.37 | 29.49 | 59.16 | 21.90 | 50.67 | 19.86 | 60.85 | 43.97 |
| | iDETR | 44.61 | 49.82 | 22.81 | 23.24 | 69.75 | 61.43 | 31.73 | 56.27 | 55.40 | 62.44 | 28.45 | 60.33 | 43.33 | 73.64 | 58.84 |
| | AT | 44.11 | 46.23 | 21.55 | 23.62 | 66.60 | 58.96 | 27.68 | 53.97 | 54.58 | 62.47 | 26.94 | 53.17 | 20.37 | 70.31 | 60.71 |
| | ZiRa | 46.44 | 48.56 | 20.34 | 19.64 | 69.47 | 60.00 | 30.09 | 56.27 | 58.12 | 65.13 | 25.83 | 55.69 | 39.50 | 72.59 | 58.55 |
| | DitHub | 46.40 | 49.19 | 21.33 | 24.34 | 69.23 | 61.99 | 42.10 | 57.08 | 58.85 | 55.24 | 24.62 | 57.58 | 33.56 | 70.99 | 62.50 |
| 5 | TFA | 26.43 | 45.76 | 21.92 | 22.30 | 67.40 | 60.72 | 30.63 | 53.56 | 46.80 | 63.60 | 26.88 | 56.26 | 28.00 | 64.28 | 52.49 |
| | iDETR | 43.51 | 51.65 | 25.69 | 25.53 | 70.42 | 62.98 | 49.98 | 50.54 | 54.85 | 64.80 | 33.24 | 57.64 | 42.36 | 76.51 | 56.92 |
| | AT | 43.67 | 47.16 | 14.63 | 24.97 | 66.56 | 64.19 | 38.85 | 42.03 | 55.49 | 65.20 | 27.48 | 52.68 | 32.21 | 71.23 | 57.54 |
| | ZiRa | 45.41 | 51.77 | 24.22 | 26.40 | 71.34 | 63.96 | 49.54 | 60.51 | 60.11 | 64.39 | 32.20 | 56.31 | 40.61 | 69.16 | 54.30 |
| | DitHub | 45.86 | 52.85 | 26.84 | 28.91 | 68.46 | 60.25 | 51.93 | 55.10 | 60.43 | 59.44 | 33.18 | 68.12 | 38.05 | 75.04 | 61.34 |
| 10 | TFA | 34.35 | 46.61 | 21.17 | 22.16 | 66.82 | 60.63 | 32.35 | 50.15 | 55.54 | 64.98 | 27.59 | 57.40 | 28.14 | 66.82 | 52.11 |
| | iDETR | 43.54 | 53.29 | 25.39 | 27.70 | 65.62 | 67.58 | 47.99 | 60.20 | 56.32 | 63.93 | 35.18 | 59.37 | 53.63 | 74.70 | 55.12 |
| | AT | 43.06 | 47.34 | 18.73 | 25.42 | 69.77 | 66.34 | 35.84 | 48.25 | 53.39 | 64.07 | 28.89 | 50.33 | 32.50 | 66.57 | 55.37 |
| | ZiRa | 46.14 | 53.20 | 25.56 | 25.96 | 71.72 | 65.28 | 48.22 | 62.81 | 61.24 | 65.47 | 35.58 | 55.73 | 46.86 | 70.97 | 56.14 |
| | DitHub | 46.54 | 54.43 | 26.19 | 30.28 | 68.11 | 65.73 | 52.82 | 56.82 | 63.36 | 61.85 | 36.68 | 66.82 | 41.46 | 76.00 | 61.48 |
| Full | TFA | 30.97 | 47.93 | 23.80 | 30.65 | 67.21 | 61.77 | 30.52 | 50.23 | 47.73 | 60.91 | 29.25 | 61.72 | 31.42 | 66.23 | 61.61 |
| | iDETR | 37.32 | 58.71 | 32.64 | 46.65 | 70.99 | 68.56 | 55.32 | 58.88 | 64.48 | 71.01 | 50.33 | 63.30 | 39.19 | 77.12 | 64.80 |
| | AT | 42.30 | 51.14 | 23.62 | 39.90 | 72.32 | 65.51 | 31.47 | 50.48 | 60.51 | 66.07 | 39.09 | 53.50 | 34.04 | 68.07 | 60.23 |
| | OW-DETR | 31.22 | 55.58 | 28.46 | 43.78 | 70.54 | 67.78 | 43.84 | 56.75 | 63.13 | 69.51 | 45.16 | 58.99 | 36.99 | 74.42 | 63.20 |
| | CL-DETR | 32.15 | 57.26 | 29.35 | 45.15 | 71.94 | 69.90 | 45.21 | 58.52 | 65.10 | 71.68 | 46.58 | 60.83 | 38.14 | 76.74 | 65.18 |
| | ZiRA | 46.26 | 57.98 | 31.76 | 47.35 | 71.77 | 64.74 | 46.53 | 62.66 | 66.39 | 71.00 | 48.48 | 63.03 | 41.44 | 76.13 | 62.44 |
| | DitHub | **47.01** | **62.19** | **34.62** | **50.65** | 70.46 | 68.56 | 49.28 | **65.57** | **69.58** | 71.10 | **56.65** | **70.88** | **52.82** | **79.30** | **68.18** |

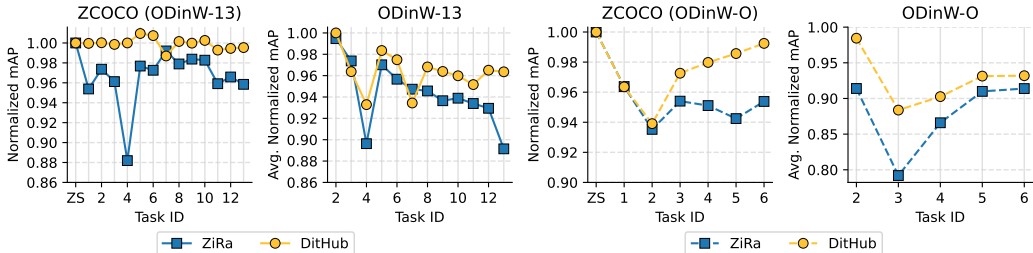

Figure 4: Normalized Average mAP as tasks progress. To assess performance degradation, we normalize the mAP for each task by dividing the current value by the original mAP recorded at the end of that task, *i.e.*, if the curve $\approx 0.90$, the model retains $90\%$ of its original performance.

for data-scarce conditions, DitHub also excels in the few-shot setting. In particular, in the five-shot and ten-shot settings, DitHub emerges as the best-performing approach, with the ten-shot scenario showing a slightly larger improvement over ZiRa (+1.23 mAP). Our method also performs well in the one-shot setting, where iDETR leads, followed closely by DitHub and ZiRa.

These findings – optimal performance in Incremental Learning, effective retention of zero-shot capabilities, and robustness in the low-data regime – clearly highlight the advantages of our modular approach.

**Performance on ODinW-O.** The ODinW-O scenario emphasizes that frequently encountered classes may reappear in subsequent tasks or at later times, such as the class "person" in common industrial and smart-city scenarios. As detailed in Section C.2, ODinW-O guarantees that each class is revisited at least once in subsequent tasks. In this benchmark, we primarily compare our approach with ZiRa, as it is the most competitive method in the standard IVLOD setting, excluding ours.

The results in Table 2 indicate that our approach secures a substantial improvement of +4.75 in mAP and a +2.08 mAP gain on ZCOCO over ZiRa. We attribute the success of our method to its class-oriented modular design, which enables selective updates to recurring concepts. This design prevents unintentional overwriting of knowledge across classes, tasks, and domains.

Table 2: mAP values on ODinW-O. Best in bold.

| Method | ZCOCO | Avg | Ae | Hw | Pv | Sd | Th | Ve |
|---|---|---|---|---|---|---|---|---|
| G-Dino | 47.41 | 53.15 | 45.12 | 67.54 | 58.11 | 25.84 | 70.40 | 51.87 |
| ZiRa | 44.43 | 57.63 | 39.92 | 68.00 | 64.90 | **46.26** | 77.26 | 49.47 |
| DitHub | **46.51** | **62.38** | **53.35** | **71.07** | **71.01** | 41.75 | **80.21** | **56.90** |

Table 3: Experts-not-Experts (EnE). Training by selecting a random expert.

| Method | ZCOCO | Avg | Ae | Aq | Co | Eg | Mu | Pa | Pv | Pi | Po | Ra | Sh | Th | Ve |
|---|---|---|---|---|---|---|---|---|---|---|---|---|---|---|---|
| G-Dino | 47.41 | 46.80 | 19.11 | 20.82 | 64.75 | 59.98 | 25.34 | 56.27 | 54.80 | 65.94 | 22.13 | 62.02 | 32.85 | 70.38 | 57.07 |
| EnE | 46.86 | 60.96 | 40.51 | 44.88 | 69.67 | 67.54 | 51.62 | 63.42 | 70.32 | 70.29 | 53.76 | 69.93 | 45.19 | 78.36 | 66.95 |
| DitHub | 47.01 | 62.19 | 34.62 | 50.65 | 70.46 | 68.56 | 49.28 | 65.57 | 69.58 | 71.10 | 56.65 | 70.88 | 52.82 | 79.30 | 68.18 |

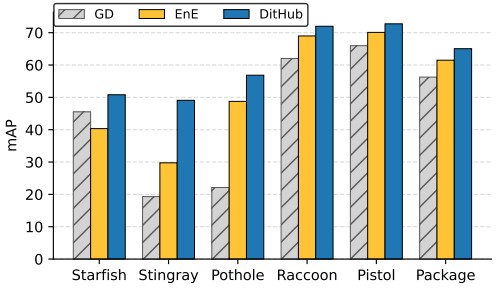

Figure 5: Performance comparison of Grounding DINO, EnE, and DitHub on rare object categories from the ODinW-13 dataset.

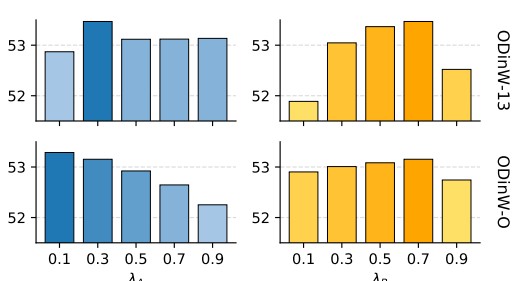

Figure 6: Sensitivity analysis on $\lambda_A$ and $\lambda_B$. Performance measured as harmonic mean of Avg and ZCOCO mAP.

**Performance analysis as tasks progress.** In Figure 4, we present an incremental analysis comparing the performance of DitHub and its strongest competitor, ZiRa, after each task. The evaluation, conducted on both ODinW-13 (left) and ODinW-O (right), allows us to assess the retention of both zero-shot and in-domain capabilities throughout the task sequence, providing a direct measure of their resistance to *forgetting* (further discussed in Section B). On ZCOCO, DitHub exhibits stable performance across tasks, with a negligible loss in zero-shot capabilities — unlike ZiRa, which shows a more pronounced decline. When evaluating performance across the sequence of downstream tasks, both methods demonstrate a decreasing trend, indicating that forgetting remains a significant challenge in Incremental Object Detection. Nevertheless, as more tasks are introduced, the performance gap widens, with DitHub consistently outperforming its competitor. This highlights the effectiveness of our modular design, which enables scalable knowledge organization and adaptation as the number of tasks grows. Moreover, the gap is still present in the ODinW-O setting introduced in this work, which models the realistic scenario where subsequent tasks may share classes. This setting, being inherently more susceptible to forgetting, further underscores the advantages of our approach.

## 5.2 Model Analysis and Ablation Studies

We examine the properties of our specialization framework and the benefits of composing class-specific experts. To this aim, we introduce an ablative approach, ***Experts-not-Experts*** (**EnE**), which removes class specialization. EnE is identical to DitHub, except during training, where it omits the stochastic strategy from Section 4.1. Instead, given an image from the current task, EnE performs the training step on one module chosen at random among all task classes. As a result, modules no longer specialize in a class but instead learn from the whole task data, including images without their designated class. Consequently, they become generalist learners rather than class-specific experts.

In Table 3, we compare DitHub with its non-specialized variant, *i.e.*, EnE. Notice that, at inference, EnE acts as an ensemble of class-agnostic modules. In contrast, DitHub employs class-specialized counterparts with an expertise on data containing the same class. This targeted specialization yields a +1.23 mAP improvement on average, with additional gains also on the ZCOCO metric. These results affirm that a system of modular experts, each with distinct knowledge, offers a promising approach for systems requiring continuous and selective adaptation.

**Evaluation on rare classes.** As discussed in Section 4.3, DitHub offers fine-grained control at inference, enabling users to tailor the system to their use case — including the option to use a single module specialized for a target class. To assess this, we conduct a targeted evaluation on rare classes from ODinW-13, activating only the corresponding target class module. Results in Figure 5 show that DitHub achieves the best results on rare classes, highlighting the benefits of class specialization.

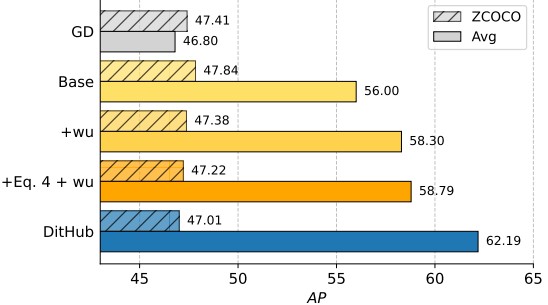

| Method | Memory (MB) | Avg (mAP) |
|---|---|---|
| DitHub $r = 1$ | $\sim 9$ | 57.04 |
| ZiRa | $\sim 18$ | 57.98 |
| DitHub $r = 2$ | $\sim 18$ | 60.26 |
| DitHub $r = 4$ | $\sim 37$ | 60.53 |
| DitHub $r = 8$ | $\sim 74$ | 61.93 |
| DitHub $r = 16$ | $\sim 147$ | 62.19 |
| G-Dino *(encoder only)* | $\sim 659$ | 46.80 |

Figure 7: *(left)* Ablation study on DitHub components, showing average performance on ODinW-13 alongside the ZCOCO metric. *(right)* Memory usage (MB) versus performance (mAP) for DitHub at varying ranks, compared against ZiRa, with performance averaged over the ODinW-13 tasks.

**Sensitivity Analysis of $\lambda$.** Varying the merging coefficients $\lambda_A$ and $\lambda_B$ offers valuable insights into our method. To this end, we define an aggregate metric as the harmonic mean of DitHub's full-shot performance, balancing specialization (Avg) and zero-shot generalization (ZCOCO). Figure 6 shows performance trends for ODinW-13 (top) and ODinW-O (bottom) as $\lambda_A$ and $\lambda_B$ in $0.2$ increments.

Considering $\lambda_A$, its role is significant while merging the expert $A$ matrices (see Equation 3). From Figure 6, lower values of $\lambda_A$ lead to better results, indicating that prioritizing past expert knowledge over the warmup module is more effective. Interestingly, while $\lambda_A = 0.3$ performs best for ODinW-13, ODinW-O exhibits a monotonic decline as $\lambda_A$ increases. The contrast reflects dataset structure: ODinW-O includes overlapping classes, emphasizing knowledge retention, whereas ODinW-13 has varied class distributions, demanding stronger integration of domain-specific warmup knowledge.

For the incremental adaptation of the shared $B$ matrix, a different trend emerges. The harmonic mean increases with $\lambda_B$, peaking at $0.7$ for both datasets. This suggests that prioritizing the knowledge in the $B^{\text{opt}}$ matrix from the current task, as shown in Equation 4, is crucial. Our intuition, supported by [30], is that the latest $B^{\text{opt}}$ matrix integrates knowledge from both earlier and recent tasks, making it more valuable than $B_{t-1}$, which is trained solely on earlier tasks.

**On Warmup and Shared $B$.** To evaluate the impact of DitHub's components, we progressively integrate them and present results in Figure 7 *(left)*. We establish the zero-shot performance of Grounding DINO (GD) as a reference. Our baseline fine-tuning procedure (Base) trains class-specific $A$ matrices and the shared $B$ matrix independently for each new task, omitting both parameter merging and the warmup phase. While this baseline suffers from complete catastrophic forgetting, it still demonstrates improved performance compared to the zero-shot Grounding DINO.

Building on this baseline, we first add the warmup phase (+wu), which yields a notable improvement by stabilizing training and facilitating the learning of rare classes through robust initialization. Subsequently, incorporating Equation 4 into the warmup-augmented strategy results in a modest improvement — this is expected, as the shared $B$ matrix constitutes only a small fraction of the total parameters. Crucially, the most substantial performance gain is achieved upon integrating Equation 3 (DitHub in Figure 7 *(left)*), highlighting the importance of retaining domain-specific knowledge within class-specific modules to support stable and effective Incremental Learning. Collectively, the synergistic combination of these three key components – the warmup phase for robust initialization, the merging strategy for $A$ matrices to retain class-specific knowledge across domains, and the shared $B$ matrix for efficiency – yields a substantial **+6.13** mAP improvement compared to the baseline model.

Finally, we note that results show a progressive decrease in the ZCOCO metric as the average mAP rises — an expected trade-off as specialization degrades zero-shot capabilities. Notably, the Base fine-tuning baseline scores higher on ZCOCO than the original Grounding DINO zero-shot model. We attribute this counter-intuitive result to catastrophic forgetting. Our full method, DitHub, is trained to be robust across the varied domains of ODinW. In contrast, the Base procedure completely forgets such broad knowledge and overfits to the new tasks. This process of forgetting diverse, out-of-distribution data results in a model that is coincidentally more aligned with the data distribution of the COCO benchmark, leading to an artificially inflated ZCOCO score.

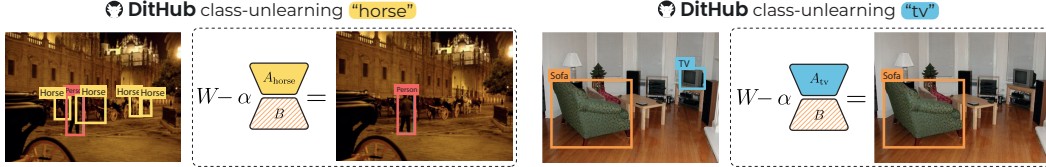

Figure 8: Qualitative results of DitHub's class unlearning. Each column compares zero-shot Grounding DINO predictions (*left*) with the output after subtracting a specific class module (*right*).

**Analysis of the Performance-Memory Tradeoff.** To study the trade-off between performance and memory, we examined how the average mAP on ODinW-13 is affected by the LoRA rank $r$, i.e., the dimensionality of the low-rank update matrices. The results, presented in Figure 7 *(right)*, demonstrate a graceful degradation in performance even as the rank is substantially reduced. Notably, a rank of $r = 2$ achieves performance within 2 mAP points of our best model ($r = 16$) while reducing the per-class parameters and memory usage by a factor of eight. We also evaluated the extreme case of $r = 1$, which collapses the adaptation matrix into a vector, and found that the performance remains remarkably strong. We attribute this robustness to our framework's granularity; since each module only encodes the knowledge of a single class, it can operate effectively with fewer parameters than approaches that aggregate information from multiple classes into one module.

In terms of memory efficiency, our method with $r = 2$ requires approximately the same memory as ZiRa while performing significantly better (**+2.28** mAP). Furthermore, our $r = 1$ variant delivers highly competitive results, scoring less than 1 mAP point below ZiRa while using only half the memory. A detailed breakdown of the per-task mAP for these varying ranks is provided in Section A.3.

**Specialized modules allow training-free unlearning.** The advantages of DitHub go beyond additive Incremental Learning. Expert modules can also be used to remove specific knowledge by simply subtracting the corresponding weights from the base model. This operation, akin to *zero-training unlearning*, has been studied in image and text categorization [15, 34]; to the best of our knowledge, we are the first to explore its application within the context of Object Detection. Such a mechanism allows the model to unlearn the detection of sensitive object categories – such as faces, license plates, or medical equipment – when required to comply with privacy regulations or data retention policies.

Specifically, we test whether class knowledge can be unlearned from the zero-shot Grounding DINO baseline by subtracting the corresponding class $c$ module from the pre-trained model: $W' = W - \alpha BA_c$, where we set $\alpha = 0.3$. Figure 8 qualitatively demonstrates DitHub's class-unlearning capability. Each column illustrates the effect of removing a specific class module. For instance, when the horse module is subtracted, the previously detected horses are no longer recognized, resulting in false negatives that indicate successful forgetting. This highlights DitHub's ability to selectively unlearn target classes while preserving knowledge of unrelated ones. A quantitative analysis is provided in Section D.2.

## 6  Conclusions

In this research, we introduced DitHub, a methodology akin to a version control system that maintains an expanding library of modules to adapt Open-Vocabulary object detectors. Our approach achieves state-of-the-art performance in the Incremental Vision-Language Object Detection scenario. The incremental nature of this domain allowed us to explore the compositional capabilities of efficient modules, marking the first such investigation in Object Detection.

In future works, we aim to further leverage modular library management to deepen our understanding of knowledge transfer between efficient modules for Object Detection. For instance, if our library contains an "axolotl" module trained on real-world images and we possess domain knowledge for the cartoon image domain, can we synthesize a module specialized in detecting cartoon "axolotls" through knowledge transfer? DitHub will provide a `fork` of this manuscript to explore such questions.

## Acknowledgments and Disclosure of Funding

Chiara Cappellino's position was funded through the project supported by the European Defence Fund (EDF) under grant agreement EDF-2022-101121405. Angelo Porrello was financially supported by the Italian Ministry for University and Research – through the ECOSISTER ECS 00000033 CUP E93C22001100001 project – and the European Commission under the Next Generation EU programme PNRR. Rita Cucchiara was financially supported by the EU Horizon project "ELIAS - European Lighthouse of AI for Sustainability" (No. 101120237).

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

## Ethics Statement

This work focuses on improving the efficiency and robustness of object detection models. All experiments were conducted on publicly available datasets that do not contain personally identifiable information. The proposed methods could, in principle, be applied to surveillance or monitoring systems; however, our goal is to advance fundamental research in detection architectures rather than enabling privacy-invasive applications. We encourage the responsible and transparent use of these techniques in compliance with data protection regulations.

## Appendix Overview

This appendix provides supplementary information to the main manuscript, organized as follows.

**Section A — Further Insights on Efficiency.** Analyzes the performance optimization of DitHub when adapting Grounding DINO.

- *Section A.1 — Blockwise Adaptation Analysis:* Justifies adapting only the encoder for DitHub based on empirical metrics.
- *Section A.2 — Employing Full LoRA for Class-Specificity:* Explores class-specific LoRA, ultimately favoring a parameter-efficient shared $B$ matrix.
- *Section A.3 — Performance-Memory Tradeo-ff Details:* Details the per-task mAP for the experiments introduced in the right panel of Figure 7.

**Section B — Further Insights on Forgetting.** Evaluates the anti-forgetting capabilities of DitHub.

- *Section B.1 — Forgetting on ODinW-13:* Presents anti-forgetting results for the ODinW-13 benchmark.
- *Section B.2 — Forgetting on ODinW-O:* Demonstrates robust anti-forgetting performance on the ODinW-O benchmark.

**Section C — Additional Benchmark Details.** Provides comprehensive information about the datasets used.

- *Section C.1 — ODinW-13 and MS COCO:* Describes the ODinW-13 dataset for specialization and MS COCO (ZCOCO) for zero-shot evaluation.
- *Section C.2 — ODinW-O:* Introduces the ODinW-O benchmark for Incremental Learning and domain shift analysis.

**Section D — Additional Results.** Presents complementary experimental results.

- *Section D.1 — Standard Deviations:* Reports standard deviations for the reproduced methods in Table 1 and Table 2.
- *Section D.2 — Further Unlearning Results:* Illustrates the modular capability of DitHub for selective knowledge removal, as shown in Figure C.
- *Section D.3 — Performance with Swin-B Backbone:* Shows the scaling capabilities of DitHub when employing a larger backbone.
- *Section D.4 — Beyond ODinW-13:* Presents results for the AODRaw benchmark.

**Section E — Implementation Details.** Outlines the experimental setup, including hardware, model configurations, and optimization specifics.

**Section F — Limitations and Broader Impact.** Discusses the current limitations of the proposed method and its broader societal implications.

**Section G — Qualitative Results.** Provides visual comparisons in Figure D that demonstrate the improved detection accuracy and robustness of DitHub.

Table A: Performance comparison across datasets when adapting only the decoder (Only Dec), both encoder and decoder (Dec/Enc), and encoder only (DitHub).

| Method | ZCOCO | Avg | Ae | Aq | Co | Eg | Mu | Pa | Pv | Pi | Po | Ra | Sh | Th | Ve |
|---|---|---|---|---|---|---|---|---|---|---|---|---|---|---|---|
| DitHub | 47.01 | 62.19 | 34.62 | 50.65 | 70.46 | 68.56 | 49.28 | 65.57 | 69.58 | 71.10 | 56.65 | 70.88 | 52.82 | 79.30 | 68.18 |
| Only Dec | 46.11 | 54.35 | 28.00 | 35.22 | 65.28 | 65.11 | 37.62 | 59.71 | 68.11 | 66.27 | 35.22 | 67.34 | 39.46 | 73.87 | 65.30 |
| Dec/Enc | 46.55 | 59.92 | 32.84 | 43.16 | 71.47 | 67.44 | 49.78 | 62.84 | 70.32 | 70.83 | 49.24 | 70.89 | 46.12 | 77.05 | 67.48 |

(a) Encoder        (b) Decoder        (c) Decoder/Encoder

Figure A: Blockwise analysis of adaptation strategies: (a) encoder-only adaptation (our selected method), (b) decoder-only adaptation, and (c) joint encoder-decoder adaptation.

# A  Further Insights on Efficiency

## A.1  Blockwise Adaptation Analysis

While Grounding DINO consists of four main components – an image backbone for visual encoding, a textual backbone for language understanding, and an encoder-decoder pair for detection – we focus our adaptation on the latter two. This choice is motivated by the need for a more detection-centric adaptation and efficiency considerations, as the textual backbone, being a BERT Base model [5], accounts for more than half of Grounding DINO 's parameters. In Table A, we present results for adapting only the encoder (Figure Aa), only the decoder (Figure Ab), and both encoder and decoder (Figure Ac). DitHub employs the first approach – adapting only the encoder – as it delivers the best performance on both ZCOCO and Avg metrics.

## A.2  Employing Full LoRA for Class-Specificity

While our methodology leverages a shared and unique $B$ matrix for all tasks, Table B presents results obtained using class-specific pairs. In this variant of DitHub, both the $A$ and $B$ low-rank matrices of LoRA undergo the same specialized optimization process: an initial warmup phase followed by dedicated specialization. In other words, the training pipeline for matrix $B$ mirrors that of matrix $A$ described in Section 4.1.

Table B: LoRA full results on ODinW-13, with DitHub results taken from Table 1 for reference.

| Method | ZCOCO | Avg |
|---|---|---|
| DitHub | 47.01 | 62.19 |
| DitHub (LoRA full) | 46.66 | 63.22 |

Applying our full set of strategies to both matrices yields a +1.03 mAP improvement in average performance across the tasks of ODinW-13, albeit at the cost of doubling the number of trainable parameters (Figure 3). Prioritizing efficiency, we opt for a lightweight implementation that trades a marginal performance gain for significantly reduced memory requirements. Notably, ZCOCO results favor our efficient approach, as a single, unspecialized $B$ matrix remains more aligned with the pretraining distribution, thereby mitigating degradation in zero-shot capabilities.

## A.3  Performance-Memory Trade-off Details

Table C details the per-task mAP for the experiments introduced in Figure 7 *(right)*.

# B  Further Insights on Forgetting

This section presents an additional analysis of the anti-forgetting properties of DitHub, complementing the study in Section 5.2 by offering an alternative perspective. We report the forgetting metric for both ODinW-13 and ODinW-O, computed for DitHub and its primary competitor, ZiRa [4]. This

Table C: Performance of DitHub at different ranks on the ODinW-13 benchmark.

| Method | ZCOCO | Avg | Ae | Aq | Co | Eg | Mu | Pa | Pv | Pi | Po | Ra | Sh | Th | Ve |
|---|---|---|---|---|---|---|---|---|---|---|---|---|---|---|---|
| DitHub $r = 1$ | 47.14 | 57.04 | 31.89 | 37.01 | 67.93 | 72.58 | 43.63 | 53.76 | 69.80 | 67.86 | 42.92 | 72.49 | 40.31 | 76.11 | 65.27 |
| DitHub $r = 2$ | 46.66 | 60.25 | 35.86 | 49.59 | 71.93 | 74.27 | 54.37 | 63.37 | 67.61 | 70.50 | 39.65 | 64.05 | 49.39 | 76.32 | 66.41 |
| DitHub $r = 4$ | 46.73 | 60.53 | 38.54 | 48.59 | 71.13 | 74.45 | 50.46 | 61.49 | 68.20 | 69.97 | 41.48 | 64.97 | 55.64 | 78.13 | 63.84 |
| DitHub $r = 8$ | 46.81 | 61.93 | 37.38 | 50.19 | 71.02 | 74.37 | 53.34 | 66.09 | 68.83 | 70.16 | 49.31 | 66.31 | 53.55 | 78.64 | 65.85 |
| DitHub $r = 16$ | 47.01 | 62.19 | 34.62 | 50.65 | 70.46 | 68.56 | 49.28 | 65.57 | 69.58 | 71.10 | 56.65 | 70.88 | 52.82 | 79.30 | 68.18 |

Table D: Forgetting metrics in terms of mAP for the ODinW-13 dataset. The numbers above each task indicate the order in which they were processed. Bold indicates best results.

| Method | Avg | 11 Ae | 10 Aq | 2 Co | 13 Eg | 12 Mu | 1 Pa | 7 Pv | 6 Pi | 3 Po | 4 Ra | 9 Sh | 8 Th | 5 Ve |
|---|---|---|---|---|---|---|---|---|---|---|---|---|---|---|
| ZiRa | -6.66 | -4.31 | -4.82 | **-1.80** | 0.00 | -2.09 | -8.71 | -5.80 | -7.76 | -12.63 | -19.19 | **-1.82** | -9.85 | -7.77 |
| DitHub | **-1.73** | **-1.56** | **-1.69** | -2.34 | 0.00 | **-1.61** | **3.62** | **-1.91** | **1.54** | **-6.94** | **-7.27** | -2.11 | **0.10** | **-2.34** |

metric is calculated using the canonical definition [3, 1]: the difference in performance on a specific task after learning the entire sequence of tasks compared to the original performance on that task immediately after its initial learning.

## B.1 Forgetting on ODinW-13

Table D presents the forgetting in terms of mAP points for the 13 tasks of ODinW-13, with the average across all tasks reported as Avg. DitHub demonstrates strong performance on both average and individual tasks, consistently outperforming its competitor by a notable margin. Given the limited class overlap within ODinW-13, the primary component susceptible to catastrophic forgetting in this analysis is the shared matrix $B$.

Notably, for some tasks, this metric is even positive for our approach. We attribute this behavior to the initialization of the $B$ component. Specifically, the metric is higher in the first learned task (Pa) [3]. While this might seem counterintuitive, the Pa dataset comprises very few instances, which means the $B$ matrix cannot undergo robust initialization. Consequently, subsequent tasks introduce greater data diversity, leading to improved performance compared to the initial one limited by scarce training data.

## B.2 Forgetting on ODinW-O

Table E reports the forgetting metric results for the ODinW-O benchmark. Unlike the previous analysis, this assessment allows us to evaluate the forgetting of DitHub's class-specific $A$ components for all classes within the dataset. Indeed, ODinW-O is specifically designed to test Incremental Learning capabilities, featuring complete class overlap across tasks. This overlap leads to catastrophic forgetting of the $A_c$ expert compo-

Table E: Forgetting metrics in terms of mAP for the ODinW-O dataset. Bold indicates best results.

| Method | Avg | 5 Ae | 6 Hw | 1 Pv | 2 Sd | 4 Th | 3 Ve |
|---|---|---|---|---|---|---|---|
| ZiRa | -3.84 | **-0.08** | 0.00 | -9.67 | -10.66 | -0.98 | -1.62 |
| DitHub | **-2.68** | -4.75 | 0.00 | **-2.74** | **-9.51** | **1.74** | **-0.79** |

nent for the same class $c$ across different domains. It's worth noting that even in this setting, which emphasizes the $A$ component, the forgetting of $B$ remains a concern. Even in this challenging context, DitHub surpasses its competitor by an appreciable margin, demonstrating difficulty only with the Ae task and highlighting the non-trivial nature of merging and preserving knowledge from the aerial domain.

## C  Additional Benchmark Details

This section provides additional details regarding the datasets utilized in our evaluation framework. We first outline the key characteristics of ODinW-13 and MS COCO, as these datasets constitute the foundation for evaluating the specialization and zero-shot capabilities of DitHub. Subsequently,

---

[3]The order in which tasks are presented to the model is random and dependent on the selected seed.

| Person | Car | Truck | Dog | Boat |
|--------|-----|-------|-----|------|

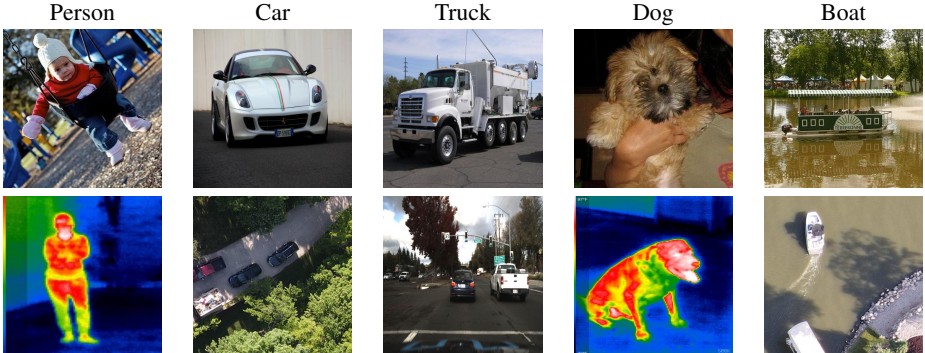

Figure B: Visualization of domain shifts for shared categories across different tasks within the ODinW-O benchmark.

we introduce and discuss the ODinW-O benchmark in greater depth, which we explicitly design to evaluate Incremental Learning performance under scenarios involving recurring categories across multiple tasks, accompanied by domain shifts. By leveraging these datasets, we establish a rigorous evaluation protocol to comprehensively assess model robustness in challenging real-world scenarios.

## C.1    ODinW-13 and MS COCO

Following [4], we assess our model specialization using the Object Detection in the Wild (ODinW-13) [20] benchmark and the MS COCO [23] dataset to verify whether the zero-shot capabilities are preserved after the sequential fine-tuning on ODinW-13.

Unlike structured and standardized datasets such as MS COCO, ODinW-13 aggregates data from thirteen sources: *Aerial Maritime Drone* (Ae), *Aquarium* (Aq), *Cottontail Rabbit* (Co), *EgoHands* (Eg), *Mushrooms* (Mu), *Packages* (Pa), *Pascal VOC* (Pv), *Pistols* (Pi), *Pothole* (Po), *Raccoon* (Ra), *Shellfish* (Sh), *Thermal Dogs and People* (Th), *Vehicles* (Ve). This comprehensive collection covers rare object categories, diverse contextual conditions, and challenging domains. Consequently, the ODinW-13 benchmark offers an ideal scenario to evaluate the capability of DitHub to sequentially adapt to heterogeneous task distributions.

Following the convention established by [4], we use the term ZCOCO to denote the zero-shot evaluation conducted on the MS COCO validation set comprising 5000 samples and consisting of annotations for 80 common object categories. The ZCOCO evaluation specifically measures the robustness and generalization capabilities of DitHub in zero-shot scenarios, after training on each successive task from ODinW-13.

## C.2    ODinW-O

We introduce a new benchmark, ODinW-O (Overlapped), designed to further evaluate the class-incremental and specialization capabilities of DitHub. Derived from ODinW-35 [19], ODinW-O retains only classes shared across at least one task (*i.e.*, overlapping classes), while removing tasks with no shared classes. This filtering is crucial because retaining classes that do not reappear in subsequent tasks would cause the benchmark to resemble the original one. Specifically, the corresponding non-recurring class modules would remain frozen, requiring no further adaptation. In contrast, ODinW-O eliminates this scenario by ensuring all retained classes appear in multiple tasks. By enforcing strictly overlapping classes, ODinW-O provides a challenging benchmark for evaluating Incremental Learning. Additionally, to rigorously evaluate the model robustness to domain shifts, we designed ODinW-O by explicitly selecting datasets that feature markedly different contexts (Figure B), enabling clear insights into how effectively the model adapts to variations in visual appearance, environmental conditions, and imaging domains. We select the following datasets, already mentioned in Section C.1, and extracted only the overlapping classes:

- **Aerial Maritime Drone**: *boat* and *car*;
- **Pascal VOC**: *boat*, *car*, *dog*, and *person*;

Table F: Standard deviations of the reproduced methods in Table 1.

| Shots | Method | ZCOCO | Avg | Ae | Aq | Co | Eg | Mu | Pa | Pv | Pi | Po | Ra | Sh | Th | Ve |
|---|---|---|---|---|---|---|---|---|---|---|---|---|---|---|---|---|
| 1 | ZiRa | ±0.90 | ±2.05 | ±1.49 | ±0.17 | ±2.16 | ±0.98 | ±0.79 | ±1.63 | ±1.33 | ±7.91 | ±3.29 | ±12.17 | ±4.45 | ±3.07 | ±3.25 |
|   | DitHub | ±0.60 | ±0.81 | ±0.95 | ±0.57 | ±2.66 | ±1.58 | ±3.72 | ±0.00 | ±1.53 | ±1.10 | ±2.90 | ±7.03 | ±3.68 | ±4.66 | ±2.46 |
| 5 | ZiRa | ±0.23 | ±0.43 | ±3.76 | ±2.15 | ±1.40 | ±1.08 | ±3.54 | ±3.74 | ±1.54 | ±4.26 | ±3.73 | ±4.13 | ±2.03 | ±2.02 | ±4.16 |
|   | DitHub | ±0.42 | ±1.27 | ±2.52 | ±0.70 | ±2.42 | ±6.74 | ±0.76 | ±3.27 | ±0.17 | ±5.49 | ±4.79 | ±2.22 | ±4.29 | ±1.81 | ±1.54 |
| 10 | ZiRa | ±0.28 | ±0.66 | ±0.44 | ±2.64 | ±2.77 | ±0.52 | ±6.43 | ±0.97 | ±1.04 | ±2.17 | ±2.04 | ±3.90 | ±1.21 | ±3.71 | ±0.87 |
|   | DitHub | ±0.20 | ±0.54 | ±2.26 | ±0.72 | ±4.23 | ±3.53 | ±1.35 | ±3.19 | ±0.30 | ±2.74 | ±0.34 | ±2.99 | ±1.01 | ±1.15 | ±1.27 |
| Full | ZiRA | ±0.44 | ±0.68 | ±2.73 | ±1.17 | ±1.39 | ±4.77 | ±7.26 | ±2.19 | ±1.33 | ±0.33 | ±4.14 | ±3.57 | ±2.44 | ±2.82 | ±3.13 |
|   | DitHub | ±0.16 | ±0.44 | ±2.39 | ±1.00 | ±2.83 | ±0.48 | ±1.34 | ±0.60 | ±0.38 | ±1.68 | ±0.76 | ±1.57 | ±2.87 | ±0.69 | ±1.01 |

- **Thermal Dogs and People**: *dog* and *person*;
- **Vehicles**: *car* and *truck*.

Additionally, we introduce two further datasets from ODinW-35 to emphasize domain variation:

- **Hard Hat Workers**: *person*;
- **Self Driving Cars**: *car* and *truck*.

Despite ODinW-O enforcing class overlap and introducing domain shifts, our model consistently achieves state-of-the-art performance by effectively refining class knowledge while adapting across diverse contexts.

## D    Additional Results

### D.1    Standard Deviations

We report the standard deviations of the reproduced methods from Table 1 and Table 2 in Table F and G, respectively, computed over three runs with different random seeds.

### D.2    Further Unlearning Results

As previously discussed in Section 5.2, our modular framework facilitates the precise removal of specific knowledge from Grounding DINO through the subtraction of corresponding experts. Figure C visually represents this process: each column depicts the mean Average Precision (mAP) of a target class after unlearning different modules (reported on the y-axis), with colors normalized column-wise to emphasize relative performance decrements. The prominent darker diagonal pattern signifies that removing a class-specific module results in the most substantial performance reduction for its associated class. This observation substantiates that our modules effectively encapsulate specialized knowledge, as their unlearning selectively impacts their respective classes.

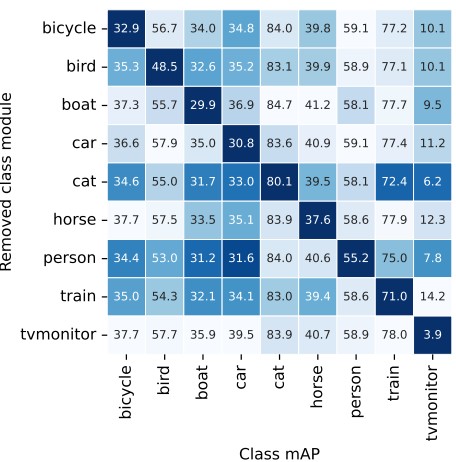

Figure C: Per-class mAP degradation after removing class modules. Darker cells indicate greater performance drops.

Furthermore, this matrix representation elucidates correlations between certain classes during the unlearning procedure. For instance, the darker shading in the "bicycle" column indicates that the "person" class exhibits the second most significant mAP drop. This correlation arises from the frequent co-occurrence of "person" instances riding bicycles, naturally linking these classes within the comprehensive unlearning pipeline.

### D.3    Performance with Swin-B Backbone

To further assess the scalability of our approach, we conducted new experiments utilizing the larger, publicly available pre-trained Grounding DINO model with a Swin-B backbone. Table H details the

Table G: Standard deviations of the reproduced methods in Table 2.

| Method | ZCOCO | Avg | Ae | Hw | Pv | Sd | Th | Ve |
|---|---|---|---|---|---|---|---|---|
| ZiRa | ±0.25 | ±0.27 | ±1.76 | ±1.14 | ±1.35 | ±0.32 | ±1.75 | ±1.76 |
| DitHub | ±0.24 | ±0.66 | ±2.39 | ±1.88 | ±0.76 | ±1.31 | ±0.46 | ±0.84 |

Table H: Performance Evaluation with the Swin-B backbone on ODinW-13.

| Method | ZCOCO | Avg | Ae | Aq | Co | Eg | Mu | Pa | Pv | Pi | Po | Ra | Sh | Th | Ve |
|---|---|---|---|---|---|---|---|---|---|---|---|---|---|---|---|
| GD ZS w/Swin-B | 56.47 | 59.82 | 33.32 | 43.26 | 72.98 | 74.90 | 37.67 | 59.14 | 70.30 | 73.56 | 48.42 | 67.30 | 62.61 | 70.69 | 64.48 |
| ZiRa w/Swin-B | 53.22 | 65.09 | 40.71 | 54.16 | 73.32 | 73.97 | 47.25 | 67.03 | 72.39 | 73.29 | 59.83 | 73.05 | 61.98 | 82.59 | 66.63 |
| DitHub w/Swin-B | 53.82 | 67.03 | 42.41 | 53.74 | 74.53 | 75.18 | 53.28 | 66.05 | 74.04 | 74.22 | 59.25 | 76.51 | 67.63 | 82.85 | 71.67 |

comparative performance of our method, DitHub, against the primary competitor, ZiRa. The findings confirm that DitHub scales effectively, maintaining its robust performance when integrated with a more powerful backbone model.

### D.4 Beyond ODinW-13

To rigorously evaluate the generalizability of our approach, we have extended our experiments to a more diverse benchmark. Our initial experimental setup necessitated a benchmark with class re-occurrence across semantically distinct domains — a characteristic not commonly found in standard long-tailed datasets.

For this purpose, we selected AODRaw [22], a challenging and recently introduced benchmark. This dataset is ideally suited for our evaluation, as it comprises 9 distinct domains with substantial class overlap, encompassing a wide range of meteorological conditions as well as both indoor and outdoor scenes. The performance analysis is presented in Table I.

## E   Implementation Details

All experiments for DitHub were conducted on a single RTX A5000 GPU with 24 GB of VRAM. Each training run, with a batch size of 2, required approximately 8 hours. Our PyTorch implementation efficiently utilized around 10 GB of VRAM during training. We employed Grounding DINO Tiny as an Open-Vocabulary object detector, which was pre-trained on the O365 [45], GoldG [16], and Cap4M [20] datasets.

For LoRA fine-tuning, we applied low-rank updates to the encoder layers with a rank of 16. The optimization was performed using AdamW [27], with a learning rate of $1 \times 10^{-3}$ and a weight decay of $1 \times 10^{-2}$. We exclusively relied on Grounding DINO's contrastive classification loss and localization loss, without introducing any additional loss functions. The final model comprised approximately 173 million parameters, with only 19.6 million LoRA parameters actively updated during training on ODinW-13. After fine-tuning on ODinW-13, only 75.65 MB of LoRA parameters were retained for storage.

## F   Limitations and Broader Impact

A limitation of the current work is that the merging strategies employed by DitHub for both class-specific experts ($A$ matrices) and the shared $B$ matrix could be further optimized. While the current solutions demonstrated effectiveness, their reliance on empirically derived coefficients highlights a potential for refinement towards strategies grounded in more robust theoretical principles. Moreover, when merging modules pertaining to the same class but originating from different tasks, incorporating explicit domain knowledge becomes crucial. We hypothesize that a mechanism explicitly accounting for domain information (*e.g*., distinguishing between RGB and thermal images) could not only enhance DitHub's performance but also facilitate the development of novel expert module merging techniques. Despite these acknowledged limitations, DitHub achieves state-of-the-art performance, underscoring the efficacy of its expert module management approach.

We propose that our method, and more broadly Modular Deep Learning, can serve as a cornerstone for fostering more accessible Artificial Intelligence. The development of methods for training

Table I: Domain-wise performance comparison on ODinW-13.

| Method | ZCOCO | Avg | fog_low_light | fog_rain | fog | indoor_low_light | indoor_normal_light | low_light_rain | low_light | normal_light | rain |
|--------|-------|-----|---------------|----------|-----|------------------|---------------------|----------------|-----------|--------------|------|
| GD ZS  | 47.40 | 15.65 | 13.32 | 19.56 | 19.42 | 15.84 | 24.77 | 11.74 | 16.28 | 5.81 | 14.11 |
| ZiRA   | 44.39 | 32.34 | 27.60 | 36.34 | 29.28 | 28.28 | 43.18 | 30.67 | 31.73 | 30.49 | 33.47 |
| DitHub | 45.23 | 34.03 | 30.30 | 35.48 | 31.40 | 31.02 | 47.77 | 31.36 | 32.88 | 30.90 | 35.12 |

and managing libraries of expert modules could enable the granular selection and deployment of knowledge onto large, otherwise computationally intractable, pre-trained architectures. Specifically, DitHub facilitates the management of modules that enable large Open-Vocabulary object detectors to recognize specific and potentially rare classes. This specificity also contributes to enhanced model controllability, potentially enabling targeted unlearning capabilities for bias mitigation and thereby advancing towards fairer deep learning systems.

# G   Qualitative Results

In Figure D, we compare zero-shot Grounding DINO (left) with ZiRA (center) and DitHub (right). Our method excels at detecting challenging objects, including small ones that other models miss. Additionally, DitHub avoids false positives (ZiRA) and false negatives (Grounding DINO), demonstrating more reliable predictions. It also correctly identifies both cars (last row) despite occlusions and distractors. This improved stability stems from per-class specialization, which mitigates hallucinations and ensures smoother adaptation to domain shifts.

| Grounding DINO | ZiRa | DitHub |
| --- | --- | --- |

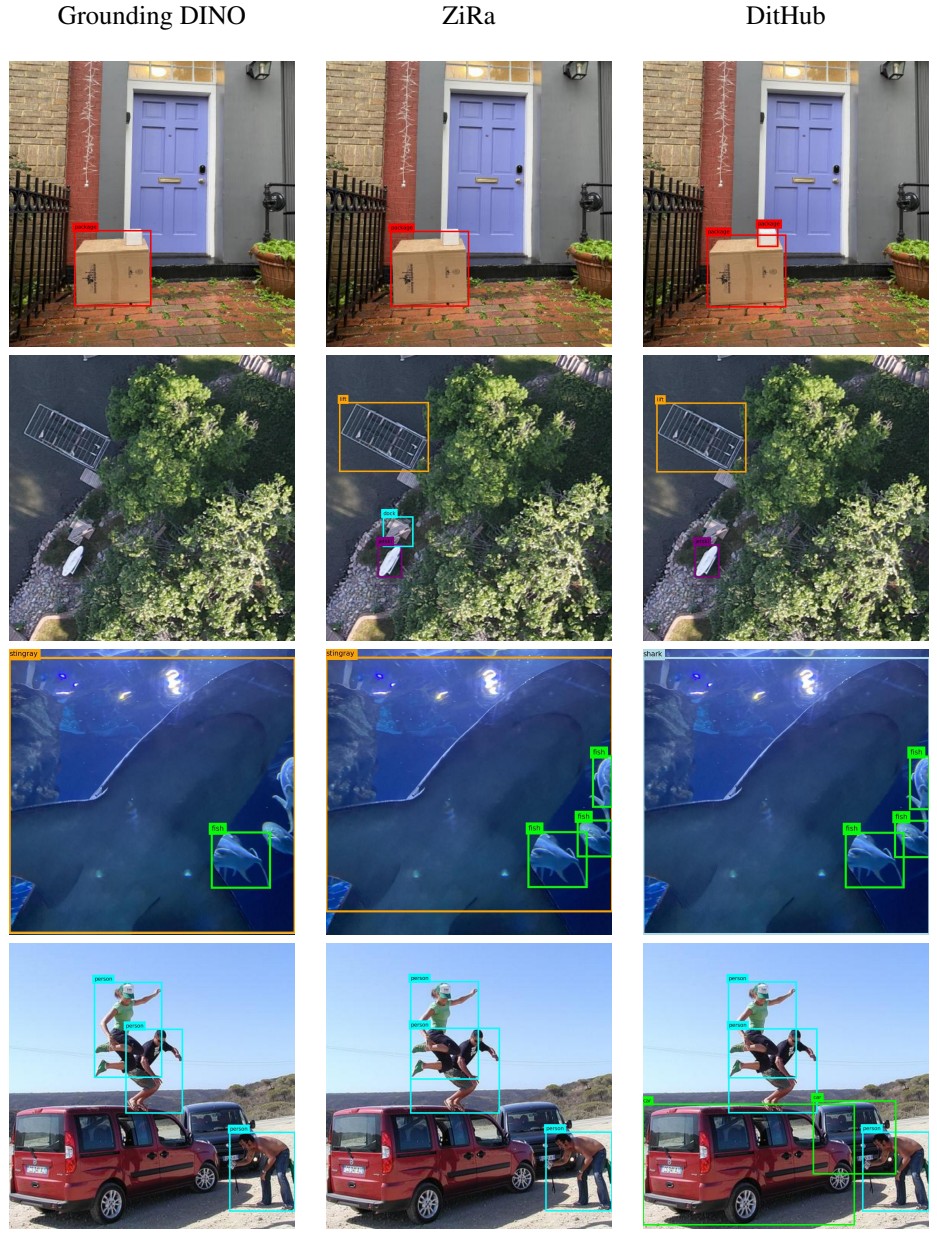

Figure D: Qualitative comparison of Grounding DINO (left), ZiRa (center), and DitHub (right).

