# OpenReview forum: "DitHub: A Modular Framework for Incremental Open-Vocabulary Object Detection"
_NeurIPS.cc/2025/Conference — NeurIPS 2025 poster_

### Official Review · Reviewer_Xs7M · 2025-06-23

**Clarity:** 2
**Significance:** 2
**Originality:** 2
**Rating:** 4
**Confidence:** 2

**Summary:**

DitHub is a framework for open-vocabulary object detection inspired by the version control system, which manages a growing library of detection modules. When encountering a new task, a class-agnostic warmup phase is adopted to provide initialization for the later specialization stage. The specialization stage will decide if a new task has common classes with previous learned tasks (the "fetch" operation) and initialize the training from there. Otherwise, it will train from the warm-up results. The authors provide detailed experiments and analysis to show the effectiveness of this method.

**Questions:**

1. If B is not shared (class-specific), what is the difference between A and B?
2. Can this method apply to other open-vocabulary tasks, like classification?
3. Following weakness 3, what is DitHub's memory usage compared to baselines?

**Ethical Concerns:**

["NO or VERY MINOR ethics concerns only"]

**Final Justification:**

The authors provide additional experiment results on semantic hierarchies and an additional dataset. Although I don't think the idea is intriguing, due to its SOTA performance and comprehensive evaluation, I think this paper can be accepted. Hence I am willing to raise my score to Borderline Accept.

**Limitations:**

Overall, I think the idea of training LoRA matrices to fine-tune Grounding-DINO for each class is a bit too simple and inefficient. Combining general and specialized information is also not original for open-vocabulary incremental learning. For other limitations, please see the weakness and question section. No potential negative societal impact.

**Paper Formatting Concerns:**

No paper formatting concerns.

**Quality:**

3

**Strengths And Weaknesses:**

Strengths:
1. The basic idea of this paper is to combine "general" information with "specialized" information. From a training aspect, it divides the training stage into "warmup" and "specialization". From a modular aspect, it uses a shared B matrix and a class-specialized A matrix. This idea is straightforward and has proven to be useful in ablation studies.
2. The introduction of the dataset ODinW-O presents realistic scenarios where classes can reoccur.
3. The experiment section is detailed and thoroughly.
4. Codes are attached.

Weaknesses:
1. I am not a big fan of the terms used in this paper. I think the idea of using terms like "merged", "fetch", and "branches" is for readers to associate the concepts of this paper with GitHub, but somehow it makes this paper more difficult to understand, as these terms do not have the same meanings as the ones used in GitHub. But it's a personal preference. Some reviewers may love it. You don't actually need to response to it in the rebuttal.
2. I think the method section should be better arranged. 4.1 focuses on two-stage training: warmup and specialization, and 4.2 focuses on memory efficiency, which proposes to share the matrix B instead of a distinct B for each class. However, B is shared has already been mentioned in 4.1. Meanwhile, Figure 3 the legend (A, B) cannot present that (A, B) is distinct for each class. Overall, I think this section is not easy to read and needs to be further polished.
3. I think training a LoRA matrix for each class is not a significant contribution and is still far from efficient and scalable.
4. The method is only tested on one benchmark ODinW-13. Seems a bit limited.

---

> ### Author Rebuttal · Authors · 2025-07-30
>
> We sincerely thank Reviewer **Xs7M** for their constructive feedback and positive assessment of our work. We particularly appreciate their acknowledgment of our straightforward and effective warm-up strategy, the thoroughness of our experimental section, the realism of our novel ODinW-O benchmark, and the value of our provided codebase—an acknowledgment we deeply value. We will now address each of their concerns in detail and incorporate all resulting adjustments into the final manuscript.
>
> **Rearrangement of the method section**
> > I think the method section should be better arranged. 4.1 focuses on two-stage training [...]
>
> We thank the reviewer for this constructive feedback. Accordingly, we will adopt the recommended rearrangement of Sections 4.1 and 4.2 and adjust the positioning of Figure 3 in the final version of the paper.
>
>
> **On the memory cost**
> > I think training a LoRA matrix for each class is not a significant contribution and is still far from efficient and scalable.
>
> We respectfully clarify that the use of per-class LoRA modules is not presented as our primary contribution, but rather as a necessary architectural choice to enable our main research goals: investigating the fine-grained compositional properties of fine-tuning modules—a direction previously unexplored in the context of open-vocabulary object detection. These properties include not only module addition, which is instrumental for incremental learning, but also module subtraction, enabling targeted unlearning. The warm-up phase, which the reviewer kindly acknowledged, is crucial to this design. It serves not only to extract general knowledge but also to initialize the class-specific modules in a shared representation space. Indeed, as suggested by prior work [1, 2], a common starting point for adaptation greatly enhances model compositionality, which is a central theme of our investigation.
>
> To address the valid concerns about efficiency and scalability, we refer the reviewer to our response to Reviewer **voji** (section "On the memory cost"), where we introduce a more **parameter-efficient variant**. This variant trains semantic-specific modules rather than class-specific ones, a strategy that significantly reduces memory overhead by grouping related classes together. Furthermore, we invite the reviewer to proceed to the following section, where we have devised additional experiments and considerations to specifically address this point.
>
> > Following weakness 3, what is DitHub's memory usage compared to baselines?
>
> Prompted by the reviewer's question, we conducted additional experiments to analyze the trade-off between performance and memory usage in our class-specific modules. Specifically, we evaluated how the average AP on ODinW-13 varies with the rank $r$ of our matrices. The results are presented in the table below.
>
> |Method|ZCOCO|Avg|Ae|Aq|Co|Eg|Mu|Pa|Pv|Pi|Po|Ra|Sh|Th|Ve
> |:---:|:---:|:---:|:---:|:---:|:---:|:---:|:---:|:---:|:---:|:---:|:---:|:---:|:---:|:---:|:---:
> |DitHub $r=1$|47.14|57.04|31.89|37.01|67.93|72.58|43.63|53.76|69.80|67.86|42.92|72.49|40.31|76.11|65.27
> |DitHub $r=2$|46.66|60.25|35.86|49.59|71.93|74.27|54.37|63.37|67.61|70.50|39.65|64.05|49.39|76.32|66.41
> |DitHub $r=4$|46.73|60.53|38.54|48.59|71.13|74.45|50.46|61.49|68.20|69.97|41.48|64.97|55.64|78.13|63.84
> |DitHub $r=8$|46.81|61.93|37.38|50.19|71.02|74.37|53.34|66.09|68.83|70.16|49.31|66.31|53.55|78.64|65.85
> |DitHub $r=16$|47.01|62.19|34.62|50.65|70.46|68.56|49.28|65.57|69.58|71.10|56.65|70.88|52.82|79.30|68.18
>
> As shown, performance degrades gracefully even as the rank substantially decreases. Notably, a rank of $r=2$ achieves a result less than 2 AP points below our best-performing model ($r=16$), while representing an 8-fold reduction in per-class parameters and memory usage. We also test the extreme case where $r=1$, which collapses the adaptation matrix into a vector, and we find the performance remains remarkably strong. We attribute this robustness to the granularity of our framework; since each module only needs to encode the knowledge of a single class, it can do so effectively with fewer parameters compared to approaches that require information aggregation from multiple classes into a single module.
>
> Below, we provide the direct memory usage comparison (in MB) against baselines, as requested by the reviewer.
>
> |Method|Memory (MB)|ODinW-13 (AP)
> |:---|:---:|:---:
> |DitHub $r=1$|~9|57.04
> |ZiRA|~18|57.98
> |DitHub $r=2$|~18|60.26
> |DitHub $r=4$|~37|60.53
> |DitHub $r=8$|~74|61.93
> |DitHub $r=16$|~147|62.19
> |GDino *(encoder only)*|~659|46.80
>
> The results show that our method with $r=2$ employs approximately the same memory as ZiRA while performing significantly better (+2.28 AP). Furthermore, our $r=1$ variant delivers surprisingly competitive results, scoring less than 1 AP point below ZiRa while using only half the memory.
>
> **Additional datasets**
> > The method is only tested on one benchmark ODinW-13. [...]
>
> Our focus on ODinW-13 was initially motivated by its suitability for testing incremental learning across domain gaps and because it is the primary benchmark used by ZiRa, the method that introduced our experimental setting.
>
> However, we agree with the reviewer's point and have now performed additional experiments on two diverse datasets. We evaluated our method on the large-vocabulary LVIS benchmark, as well as on AODRaw, a recent and challenging benchmark encompassing different meteorological and environmental conditions. We kindly refer the reviewer to our detailed discussion and results in the response to Reviewer **voji** (section Additional datasets).
>
> **Other questions**
> > If $B$ is not shared (class-specific), what is the difference between $A$ and $B$?
>
> Indeed, as the reviewer suggested, when matrix $B$ is not shared, it is also class-specific. In this configuration, we do not enforce any distinct roles upon $A$ and $B$. Their purpose becomes a collective one: to jointly learn a low-rank matrix ($\Delta W=BA$) that optimally adapts the model's behavior for that class. The empirical results for this configuration are reported in Appendix A.2.
>
> > Can this method apply to other open-vocabulary tasks, like classification?
>
> Yes, we believe the core principles of our method are indeed applicable to other open-vocabulary tasks, such as classification. For instance, one could maintain a library of modules to incrementally adapt the vision or text encoder of a model like CLIP [3] for specialized zero-shot classification tasks (e.g., adapting it to a fine-grained domain like "species of birds" or "models of cars").
>
> However, we chose to focus on Object Detection as we believe it is a more challenging task and, therefore, better suited to studying the compositional properties of specialized modules (e.g., composing modules to detect multiple, distinct classes in a single image).
>
> [1] Mirzadeh, Seyed Iman, et al. "Linear Mode Connectivity in Multitask and Continual Learning." ICLR 2021.
>
> [2] Qin, Yujia, et al. "Exploring Mode Connectivity for Pre-trained Language Models." Conference on Empirical Methods in Natural Language Processing 2022.
>
> [3] Radford, Alec, et al. "Learning transferable visual models from natural language supervision." ICML 2021.

---

> > ### Comment · Reviewer_Xs7M · 2025-08-04
> >
> > Thanks to the authors for the time and effort they put into the rebuttal. I think the new experiments on semantic hierarchies are interesting, and the experiments on the additional dataset provided further support for the effectiveness of this method. My only concern is that the implementation (per-class LoRA) is simple and not particularly intriguing, despite the acceptable memory cost. As for the authors' claim that the main contribution is the research goal of investigating the fine-grained compositional properties of fine-tuning modules, then different approaches of achieving this goal should be explored (e.g., additional MLP layers). Nevertheless, due to the comprehensiveness of the experiments, I would raise my score to Borderline accept and encourage the author to include all new experiments in the revised version.

---

> > > ### Author Response · Authors · 2025-08-04
> > >
> > > We sincerely thank you for raising your rating and for the valuable time you invested in our manuscript. We will build on your feedback and are committed to including all the additional experiments and results in the final version of our work.

---

### Official Review · Reviewer_i82i · 2025-07-02

**Clarity:** 3
**Significance:** 3
**Originality:** 3
**Rating:** 4
**Confidence:** 3

**Summary:**

This paper introduces DitHub, a modular system for incremental open-vocabulary object detection. Inspired by version control systems, DitHub manages a growing library of lightweight adaptation modules, allowing objects to be added or re-adapted efficiently as new tasks or domains are encountered. A key innovation is the decoupling of the adaptation process into a warmup phase (providing class-agnostic initialization) and a subsequent class-specific specialization phase. The framework is instantiated using LoRA modules and demonstrates state-of-the-art performance on both the ODinW-13 benchmark and the new ODinW-O (which tests recurring class adaptation). The authors systematically analyze memory, performance degradation, rare category adaptation, and unlearning capabilities, and include ablation and sensitivity studies.

**Questions:**

Please refer to weakness.

**Ethical Concerns:**

["NO or VERY MINOR ethics concerns only"]

**Final Justification:**

I would like to keep my original rating.

**Limitations:**

Yes

**Quality:**

3

**Strengths And Weaknesses:**

### Strengths
- DitHub is a thoughtfully constructed modular adaptation framework, directly addressing the need for selective, composable, and revisable adaptation in open-vocabulary object detection
- Figure 3 directly compares the memory requirements of DitHub's modular scheme against naive class-specific adaptation, and clearly demonstrates the practical benefit of parameter sharing.


### Weaknesses
- In Table 2, comparison on ODinW-O is limited to ZiRa and G-Dino only. It is acknowledged that ODinW-O is new, but running additional strong baselines (such as dDETR or AT) would better contextualize claims of superiority and strengthen the evidence for state-of-the-art claims on this new task.
- Although the shared $B$ matrix strategy (Figure 3) is an improvement, as the number of classes scales up (e.g., hundreds or thousands), even linear scaling could raise practical concerns, especially on limited-resource deployments. A discussion is warranted already in the main text, indicating under what conditions DitHub's memory profile could become a bottleneck and whether further compression/merging strategies are feasible.
- The mechanism for merging expert modules when encountering recurring classes (Sec. 4.1, Eq. 2) is motivated and evaluated, but deeper analysis of potential information loss/conflict when merging across heavily divergent domains is lacking. It will strengthen the work to directly visualize, for example, failure cases or class confusion arising from merging experts trained on very distinct domains.

---

> ### Author Rebuttal · Authors · 2025-07-30
>
> We sincerely thank Reviewer **i82i** for their constructive feedback and for acknowledging the strengths of our work. We are particularly grateful for the reviewer's recognition that our methodology successfully addresses the critical need for selective, composable, and revisable adaptation—our primary objective—and for their acknowledgment of our model's memory efficiency over naive approaches.
>
> We will now address each of the reviewer's concerns in detail, and all corresponding revisions will be integrated into the final manuscript.
>
> **Additional baselines on ODinW-O**
> > In Table 2, comparison on ODinW-O is limited to ZiRa and G-Dino only. [...]
>
> As requested, we have expanded our comparison on the ODinW-O benchmark to include the results for **iDETR** and **AT**. For completeness and ease of comparison, the updated table also retains G-Dino and ZiRa and includes our method, DitHub. This revised table will replace the original Table 2 in the final manuscript.
>
> ||ZCOCO|Avg|Ae|Hw|Pv|Sd|Th|Ve
> |---|---|---|---|---|---|---|---|---
> |GDino|47.41|53.15|45.12|67.54|58.11|25.84|70.40|51.87
> |iDETR|37.42|59.02|49.69|67.76|67.72|35.95|78.44|54.58
> |AT|44.94|56.89|46.89|65.79|67.28|30.56|75.98|54.83
> |ZiRa|44.43|57.64|39.92|68.00|64.90|46.26|77.26|49.47
> |DitHub|46.51|62.38|53.35|71.07|71.01|41.75|80.21|56.90
>
> The expanded results in the table are consistent with our findings in Table 1. While iDETR outperforms ZiRa on the average metric (Avg), it demonstrates weaker performance in the zero-shot scenario (ZCOCO). Conversely, AT achieves a lower average score but is more effective at preserving zero-shot capabilities, yielding results comparable to ZiRa. Our method, DitHub, confirms its state-of-the-art performance by a consistent margin.
>
> **On the memory cost**
> > [...] DitHub's memory profile could become a bottleneck [...]
>
> We thank the reviewer for acknowledging both the existing discussion on potential memory bottlenecks within the original manuscript and our efforts to enhance memory efficiency over naive class-specificity.
>
> For a more comprehensive analysis, including additional experimental studies on this topic, we kindly refer the reviewer to our detailed responses to Reviewer **voji** and Reviewer **Xs7M** ("On the memory cost" sections).
>
> **Analysis on divergent domains**
> > [...] deeper analysis of potential information loss/conflict when merging across heavily divergent domains [...]
>
> We agree that investigating the effects of merging modules from heavily divergent or semantically conflicting domains is a valuable research direction, which we also highlighted in the conclusions of our original manuscript. Prompted by the reviewer's feedback, we designed a new experiment to shed further light on this matter.
>
> The experiment focuses on the class "Person" and compares two merging scenarios. In the first scenario ("Similar Domains"), we merge two class-specific modules ($A_{class}$) that were both trained on standard RGB image datasets (e.g., Pascal VOC, Pv, and Hardhat Workers, Hw). In the second scenario ("Divergent Domains"), we merge one module trained on a standard RGB dataset (Pv) with another trained on a heavily divergent dataset—the Thermal (Th) dataset. We then evaluate the zero-shot performance of the merged modules from both scenarios on the COCO validation set. The results are presented below.
>
> |Merged Domain|ZCOCO|Pv|Th|Hw
> |:---:|:---:|:---:|:---:|:---:
> |Similar|**59.89**|67.00|-|68.24
> |Divergent|**57.64**|67.00|72.92|-
>
> We repeat the same experiment for the class "Car," using the Aerial (Ae) dataset as the divergent domain and the Vehicles (Ve) dataset as the similar one. The results are presented in the following table.
>
> |Merged Domain|ZCOCO|Pv|Ae|Ve
> |:---:|:---:|:---:|:---:|:---:
> |Similar|**48.21**|66.51|-|61.01
> |Divergent|**47.66**|66.51|72.77|-
>
> The results show that merging modules from divergent domains is, in general, more challenging than merging modules from similar ones. As the results for the "Person" class demonstrate, merging modules from **similar domains** yields a higher ZCOCO score (59.89) compared to merging from **divergent domains** (57.64). This modest drop suggests that a minor degree of negative interference may occur when merging modules from significantly different data distributions, as the reviewer hypothesized. For the "Car" class, the drop in performance is less pronounced, suggesting that the Aerial domain conflicts less with standard RGB images than the Thermal domain does.

---

> > ### Comment · Reviewer_i82i · 2025-08-05
> >
> > Thanks for detailed response. After reading other reviews, I decide to keep my original rating

---

### Official Review · Reviewer_voji · 2025-07-12

**Clarity:** 2
**Significance:** 1
**Originality:** 3
**Rating:** 5
**Confidence:** 3

**Summary:**

DitHub is a modular framework for Incremental Open-Vocabulary Object Detection, which proposes (1) a two-stage LoRA fine-tuning recipe for efficient adaptation, (2) to manage class-specific modules in a Version Control Systems manner using `branch`, `fetch`, and `merge` operations, and (3) sharing a part of weights for MemoryEfficiency. The method is evaluated on ODinW-13 and ODinW-O with Grounding DINO and demonstrates superior performance to the baseline.

**Questions:**

- As the number of categories grows, an important question arises: how can the $A$ matrices be efficiently managed, particularly in scenarios where classes exhibit clear hierarchical or semantic dependencies?

- More experiments are needed.

**Ethical Concerns:**

["NO or VERY MINOR ethics concerns only"]

**Final Justification:**

With the additional results and analysis, the authors have addressed most of my concerns, making the work overall novel and solid.

**Limitations:**

See Weaknesses and Questions.

**Paper Formatting Concerns:**

None.

**Quality:**

2

**Strengths And Weaknesses:**

- Strengths

  - In general this paper is well-written and easy to follow.
  - Achieving new state-of-the-art results in open-vocabulary settings is a substantial contribution.
  - Ablation results are detailed and show reasonable analysis of different model components.

- Weaknesses

  - In some scenarios, maintaining a separate matrix for each category can be expensive. For instance, in large-scale detection datasets like V3Det [1], which contains 13,029 categories, training and managing such a vast number of weights leads significant computational and storage costs.
  - In Fig 8, paper only reports the average performance across the tasks of ODinW-13. The ZCOCO metric is also essential, as it provides insight into how different design choices affect the forgetting after fine-tuning.
  - The paper evaluates the proposed method in only a single setting (Grounding DINO), without validation across different models or scales. The authors are encouraged to conduct experiments with diverse open-vocabulary detectors and model sizes to demonstrate the broader applicability of the approach.
  - While the paper proposes a new benchmark (ODinW-O) to test class recurrence, most experiments are still focused on ODinW-13. Additional experiments on other open-vocabulary or long-tailed benchmarks ( e.g., LVIS [2], V3Det [1], or Objects365 [3] ) would strengthen generalization claims.

[1] Wang, Jiaqi, et al. "V3det: Vast vocabulary visual detection dataset." Proceedings of the IEEE/CVF International Conference on Computer Vision. 2023.

[2] Gupta, Agrim, Piotr Dollar, and Ross Girshick. "Lvis: A dataset for large vocabulary instance segmentation." Proceedings of the IEEE/CVF conference on computer vision and pattern recognition. 2019.

[3] Shao, Shuai, et al. "Objects365: A large-scale, high-quality dataset for object detection." Proceedings of the IEEE/CVF international conference on computer vision. 2019.

---

> ### Author Rebuttal · Authors · 2025-07-30
>
> We sincerely thank Reviewer **voji** for their valuable feedback, for finding our manuscript to be well-written, and for recognizing our method's state-of-the-art performance as a substantial contribution. We also appreciate their acknowledgment of our detailed and analytically reasonable ablation studies. We will now address the reviewer's concerns in detail and incorporate all resulting adjustments into the final manuscript.
>
> **On the memory cost**
>
> > In some scenarios, maintaining a separate matrix for each category can be expensive. [...]
>
> While the memory cost can be a concern in some scenarios, our work adopts the exact experimental setting from [1] to ensure fair comparison and build upon established benchmarks. This framework focuses on enhancing a large, pre-trained model with a smaller set of specialized or rare categories (e.g., "cottontail-rabbit" or "pothole" from the ODinW-13 benchmark) or handling significant domain shifts (e.g., RGB to thermal), rather than on adding thousands of new classes.
>
> Within this established setting, our use of class-specific modules is a deliberate design choice, as it is fundamental to our core contributions: exploring the composability of learned concepts and fine-grained unlearning. An architecture without this modularity would undermine our ability to investigate these properties.
>
> To ensure this modularity remains efficient, we implement each module with only a single low-rank projection matrix ($A$, with dimensions $r \times d$, where the rank $r \ll d$). To further address the reviewer's valid concern, we have conducted new experiments specifically optimizing the memory consumption of our approach. These are detailed below and in our answer to Reviewer **Xs7M** (in the section "On the memory cost").
>
> > [...] how can the $A$ matrices be efficiently managed, particularly in scenarios where classes exhibit clear hierarchical or semantic dependencies?
>
> To address the reviewer's insightful question, we designed a new experiment to evaluate the management of $A$ matrices based on **semantic hierarchies**. Following the DitHub pipeline, instead of training a specific $A$ component for each class, we define three macro-categories: *animals*, *vehicles*, and *objects*. When a class from one of these groups is encountered during training, the corresponding group matrix (e.g., $A_{animals}$) is updated.
>
> This methodology allows us to study the semantic relationships within our learned modules and to analyze how individual class concepts can coexist within a single, hierarchically superior representation. The class-to-group mappings are as follows:
>
> group|classes
> -|-
> animals|c-rabbit, raccoon, dog, cat, cow, sheep, horse, bird
> vehicles|car, bus, motorcycle, airplane, boat, train, truck, ambulance, bicycle
> objects|pistol, package, chair, dining table, tv, couch, potted plant
>
> The results, comparing the standard DitHub (per-class $A$) with the semantic aggregation approach, are reported below.
>
> &nbsp;|Avg|c-rabbit|raccoon|dog|cat|cow|sheep|horse|bird
> -|-|:-:|:-:|-|-|-|-|-|-
> DitHub|**70.51**|71.26|69.04|84.31|84.13|67.72|62.21|66.11|59.29
> $A_{animals}$|**68.08**|68.18|66.42|83.93|82.92|65.04|60.95|63.30|51.91
>
> &nbsp;|Avg|car|bus|motorcycle|airplane|boat|train|truck|ambulance|bicycle
> -|-|-|-|:-:|:-:|-|-|-|:-:|-
> DitHub|**62.79**|55.74|76.58|58.92|77.90|45.67|78.17|38.42|85.55|48.16
> $A_{vehicles}$|**58.47**|54.04|74.47|52.96|74.19|35.55|74.33|35.90|80.12|44.63
>
> &nbsp;|Avg|pistol|package|chair|dining table|tv|couch|potted plant
> -|:-:|:-:|:-:|:-:|:-:|:-:|:-:|:-:
> DitHub|**49.27**|71.23|65.08|38.93|21.24|62.90|52.14|33.38
> $A_{objects}$|**44.55**|62.90|58.99|34.89|15.92|60.32|49.10|29.75
>
> The results are highly encouraging. The experiment confirms that a single group matrix can learn a robust, shared representation for an entire semantic category. We found that groups with high semantic relation (e.g., *animals*) are more composable than more disparate ones (e.g., *objects*), which we attribute to the semantic similarity between their constituent classes (e.g., *cat* and *dog* vs. *pistol* and *couch*).
>
> Crucially, this approach demonstrates a significant gain in efficiency. By reducing the number of trainable $A$ matrices from 24 to just 3, we observe only a minor impact on accuracy. This confirms that our modules can effectively leverage semantic dependencies, as the reviewer correctly hypothesized.
>
> **Additional metrics**
>
> > In Fig 8, paper only reports the average performance across the tasks of ODinW-13. The ZCOCO metric is also essential [...]
>
> We thank the reviewer for this suggestion. As requested, we now report the **Zero-Shot COCO** (ZCOCO) metric along with the Avg AP scores across tasks.
>
> &nbsp;|ZCOCO|Avg
> -|-|-
> GD ZS|47.41|46.80
> `Base`|47.84|56.00
> `w/ WU`|47.38|58.30
> `w/ Eq.3+WU`|47.22|58.79
> DitHub|47.01|62.19
>
> The results show a progressive decrease in the ZCOCO metric as the average AP across tasks rises, which is an expected trade-off as specialization degrades zero-shot capabilities.
>
> Notably, the `Base` fine-tuning baseline scores higher on ZCOCO than the original Grounding DINO zero-shot model. We attribute this counter-intuitive result to catastrophic forgetting. Our full method, DitHub, is trained to be robust across the varied domains of ODinW. In contrast, the `Base` procedure completely forgets such broad knowledge and overfits to the new tasks. This process of forgetting diverse, out-of-distribution data results in a model that is more aligned with the data distribution of the COCO benchmark, leading to an artificially inflated ZCOCO score.
>
> **Backbone diversity**
>
> > The paper evaluates the proposed method in only a single setting (Grounding DINO), without validation across different models or scales. [...]
>
> To address the concern regarding model scales, we have conducted new experiments using the larger, publicly available pre-trained version of Grounding DINO with a **Swin-B** backbone. The table below presents these new results for our method, DitHub, and the primary competitor, ZiRa.
>
> &nbsp;|ZCOCO|Avg|Ae|Aq|Co|Eg|Mu|Pa|Pv|Pi|Po|Ra|Sh|Th|Ve
> -|-|-|-|-|-|-|-|-|-|-|-|-|-|-|-
> GD ZS w/Swin-B|56.47|59.82|33.32|43.26|72.98|74.90|37.67|59.14|70.30|73.56|48.42|67.30|62.61|70.69|64.48
> ZiRa w/Swin-B|53.22|65.09|40.71|54.16|73.32|73.97|47.25|67.03|72.39|73.29|59.83|73.05|61.98|82.59|66.63
> DitHub w/Swin-B|53.82|67.03|42.41|53.74|74.53|75.18|53.28|66.05|74.04|74.22|59.25|76.51|67.63|82.85|71.67
>
> These results confirm that our method, DitHub, scales effectively and maintains its strong performance when integrated with a larger and more powerful backbone model.
>
> Regarding model diversity, our decision to focus on Grounding DINO was twofold. Firstly, while other Detector with large-scale pretraining on both Detection and Grounding data, like GLIP [2] exist, Grounding DINO [3] is its state-of-the-art successor, and a full adaptation of our method to GLIP would be infeasible within the rebuttal period. Secondly, and crucially, Grounding DINO is the foundational model used in the work that established our experimental setting [1], making it the most appropriate choice for a direct and fair comparison. For these reasons, we believe our in-depth evaluation on this single, highly challenging model provides a strong and valid test of our method's contributions.
>
> **Additional datasets**
>
> > [...] Additional experiments on other open-vocabulary or long-tailed benchmarks [...]
>
> To address the reviewer's concern and better demonstrate the generalizability of our approach, we have conducted new experiments on two additional, diverse benchmarks. We excluded datasets like Object365, as it was part of the pre-training corpus for the Grounding DINO backbone and would not be a valid test set. Furthermore, our original benchmark choice was guided by our experimental setting, which specifically requires **class re-occurrence** across semantically distinct domains. This property is not typically found in standard long-tailed benchmarks, as they are generally not structured with overlapping classes across divergent domains.
>
> First, we present results on the large-vocabulary dataset **LVIS**. To make this evaluation computationally feasible while retaining the complete set of 1203 categories, we adopt an established procedure from the literature [4] that uses a 10% subset of the annotations.
>
> &nbsp;|ZCOCO|LVIS (AP)
> -|-|-
> GD ZS|47.41|15.20
> ZiRA|46.67| 22.31
> Dithub|46.91| 25.51
>
> Additionally, we evaluated our method on **AODRaw** [5], a challenging benchmark recently introduced at CVPR 2025. This dataset is particularly well-suited to our setting as it features 9 distinct domains with significant class overlap, spanning varied meteorological conditions as well as indoor and outdoor scenes. The results are reported in the following table.
>
> &nbsp;|ZCOCO|Avg|fog_low_light|fog_rain|fog|indoor_low_light|indoor_normal_light|low_light_rain|low_light|normal_light|rain
> :-:|:-:|:-:|:-:|:-:|:-:|:-:|:-:|:-:|:-:|:-:|:-:
> GD ZS |47.40|15.65|13.32|19.56|19.42|15.84|24.77|11.74|16.28|5.81|14.11
> ZiRA|44.39|32.34|27.60|36.34|29.28|28.28|43.18|30.67|31.73|30.49|33.47
> Dithub|45.23|34.03|30.30|35.48|31.40|31.02|47.77|31.36|32.88|30.90|35.12
>
> These additional results on both LVIS and the challenging AODRaw benchmark provide substantial new evidence for the robustness and generalizability of our proposed method beyond the primary ODinW-13 setting.
>
> [1] Deng, Jieren, et al. "Zero-shot generalizable incremental learning for vision-language object detection." NeurIPS 2024.
>
> [2] Li, Liunian Harold, et al. "Grounded language-image pre-training." CVPR 2022.
>
> [3] Liu, Shilong, et al. "Grounding dino: Marrying dino with grounded pre-training for open-set object detection." ECCV 2024.
>
> [4] Kamath, Aishwarya, et al. "Mdetr-modulated detection for end-to-end multi-modal understanding." ICCV 2021.
>
> [5] Li, Zhong-Yu, et al. "Towards raw object detection in diverse conditions." CVPR 2025.

---

> > ### Comment · Reviewer_voji · 2025-08-01
> >
> > Thanks to the authors for the answers and additional results! The new experiment of semantic hierarchies is interesting, and a GroudingDINO zero-shot baseline is needed for comparison.

---

> > > ### Author Response · Authors · 2025-08-01
> > >
> > > Thank you for your positive feedback! We are glad to hear that you found the new results on semantic hierarchies interesting.
> > >
> > > As requested, we now provide the results for the Grounding DINO zero-shot baseline:
> > >
> > > | | Avg | c-rabbit | raccoon | dog | cat | cow | sheep | horse | bird |
> > > | :----------------------------: | :---: | :---: | :---: | :---: | :---: | :---: | :---: | :---: | :---: |
> > > | GD ZS | **63.47** | 64.75 | 62.02 | 76.21 | 80.73 | 62.97 | 52.62 | 62.24 | 46.24 |
> > > | DitHub | **70.51** | 71.26 | 69.04 | 84.31 | 84.13 | 67.72 | 62.21 | 66.11 | 59.29 |
> > > | $A_{animals}$ | **68.08** | 68.18 | 66.42 | 83.93 | 82.92 | 65.04 | 60.95 | 63.30 | 51.91 |
> > >
> > > | | Avg | car | bus | motorcycle | airplane | boat | train | truck | ambulance | bicycle |
> > > | :----------------------------: | :---: | :---: | :---: | :---: | :---: | :---: | :---: | :---: | :---: | :---: |
> > > | GD ZS | **56.43** | 51.33 | 74.56 | 50.68 | 69.94 | 35.56 | 74.58 | 36.33 | 79.15 | 35.72 |
> > > | DitHub | **62.79** | 55.74 | 76.58 | 58.92 | 77.90 | 45.67 | 78.17 | 38.42 | 85.55 | 48.16 |
> > > | $A_{vehicles}$ | **58.47** | 54.04 | 74.47 | 52.96 | 74.19 | 35.55 | 74.33 | 35.90 | 80.12 | 44.63 |
> > >
> > > | | Avg | pistol | package | chair | dining table | tv | couch | potted plant |
> > > | :----------------------------: | :---: | :---: | :---: | :---: | :---: | :---: | :---: | :---: |
> > > | GD ZS | **43.05** | 62.98 | 56.27 | 34.16 | 9.50 | 60.59 | 49.91 | 27.93 |
> > > | DitHub | **49.27** | 71.23 | 65.08 | 38.93 | 21.24 | 62.90 | 52.14 | 33.38 |
> > > | $A_{objects}$ | **44.55** | 62.90 | 58.99 | 34.89 | 15.92 | 60.32 | 49.10 | 29.75 |
> > >
> > > As shown in the table, both DitHub and especially its variant leveraging hierarchical grouping and macro-categories outperform the zero-shot baseline. This indicates that our modular approach remains effective even when categories are grouped according to semantic criteria.
> > >
> > > Furthermore, the results reinforce a key observation from our previous response: the performance of a hierarchical module directly correlates with the semantic similarity of the concepts it represents. For instance, the performance gain of the $A_{animals}$ module over the Grounding DINO zero-shot baseline is larger than the gains achieved by the $A_{vehicles}$ and, most notably, the $A_{objects}$ modules. This finding is consistent with the fact that the *objects* macro-category encompasses the most semantically diverse concepts, whereas the *animals* group is the most coherent.
> > >
> > > We appreciate your thoughtful comments and suggestions. We remain at your disposal for any further questions, clarifications, or suggestions that could help improve the final version of the paper.

---

> > > > ### Comment · Reviewer_voji · 2025-08-02
> > > >
> > > > Thanks! With the additional results and analysis, the authors have addressed most of my concerns, making this work overall novel and solid. Based on this, I increase the rating of this paper to "5: Accept".

---

> > > > > ### Author Response · Authors · 2025-08-02
> > > > >
> > > > > We sincerely thank you for raising your rating and for your thoughtful feedback on our work. As committed, the final version of our manuscript will incorporate all the new results.

---

### Decision · Program_Chairs · 2025-09-17

**Decision:**

Accept (poster)

**Comment:**

The paper received all positive reviews, leading to a final acceptance recommendation.